# Volitional exercise elicits physiological and molecular improvements in the severe D2.mdx mouse model of Duchenne muscular dystrophy

Stephanie R. Mattina[1], Sean Y. Ng[1], Andrew I. Mikhail[1], Derek W. Stouth[1], Cora E. Jornacion[1], Irena A. Rebalka[2], Thomas J. Hawke[2] and Vladimir Ljubicic[1]

[1] *Department of Kinesiology, McMaster University, Hamilton, Ontario, Canada*
[2] *Department of Pathology and Molecular Medicine, McMaster University, Hamilton, Ontario, Canada*

Handling Editors: Richard Carson & Bruno Grassi

The peer review history is available in the Supporting information section of this article (https://doi.org/10.1113/JP286768#support-information-section).

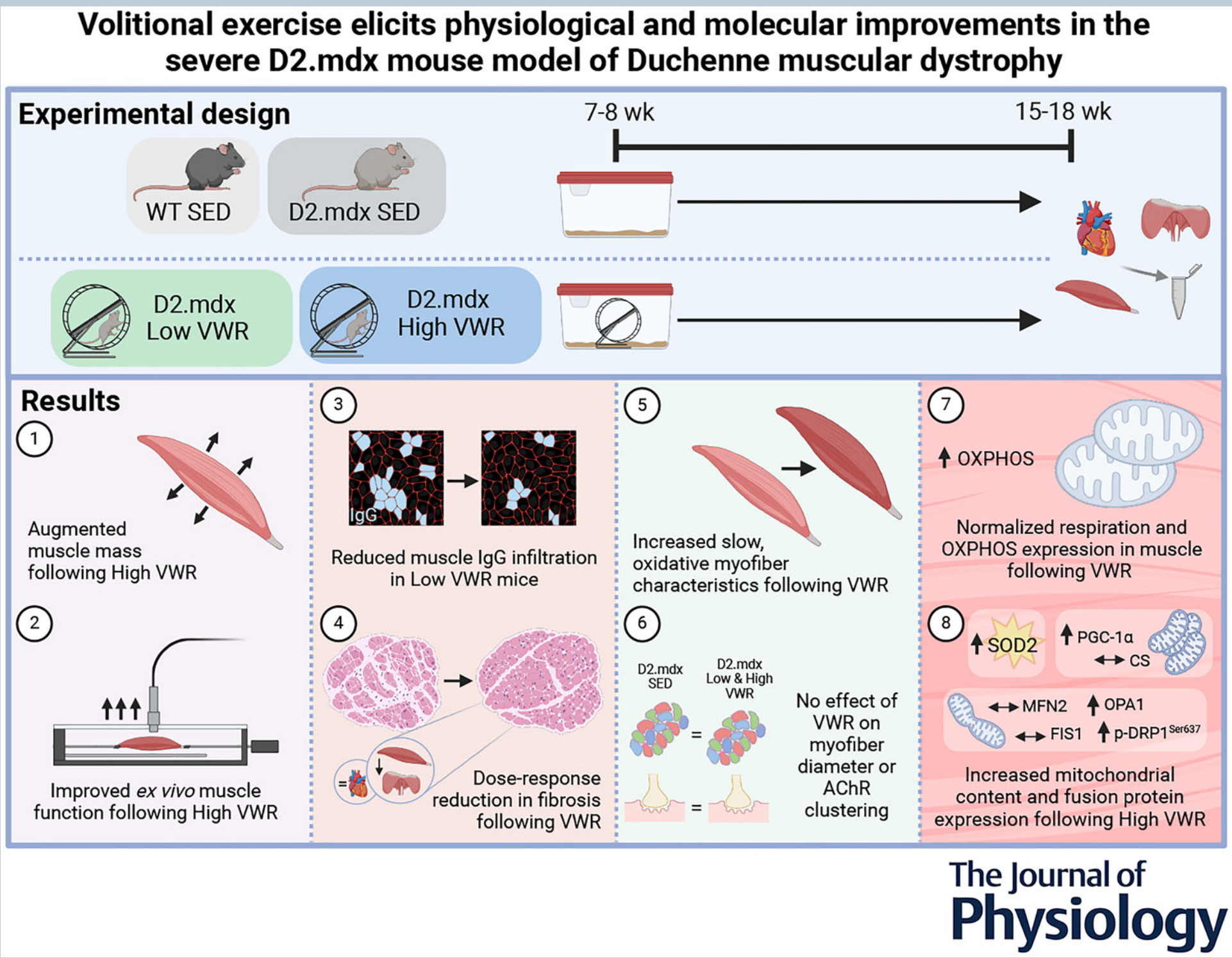


**Abstract figure legend** This study investigated the effects of volitional exercise on muscle health in the more severe D2.mdx model of Duchenne muscular dystrophy (DMD). We showed that 8–10 weeks of a relatively high volume of voluntary wheel running (VWR) in D2.mdx animals augmented select muscle mass and normalized *ex vivo* muscle force compared to sedentary counterparts. Notably, we observed a reduction in IgG infiltration within the muscle of Low VWR animals, and a dose–response decline in fibrosis in dystrophic animals that underwent VWR. Furthermore, we identified a shift towards a more slow, oxidative myofiber type, with no change in diameter or acetylcholine receptor (AChR) clustering following VWR. Finally these changes coincided with normalized mitochondrial respiration, as well as increased mitochondrial content and fusion protein content following High VWR, compared to genotype-matched sedentary controls.

**Abstract**   Duchenne muscular dystrophy (DMD) is characterized by progressive muscle wasting and weakness. Prescribed moderate exercise in patients is beneficial, but concerns remain due to the vulnerability of dystrophic muscle to damage. Voluntary wheel running (VWR) is a self-regulated form of exercise that improves muscle health in the typical C57.mdx mouse model of DMD. The purpose of the current study was to investigate the impact of VWR in more severe and clinically relevant D2.mdx mice. Male D2.mdx animals were assigned to a sedentary (D2.mdx SED) or VWR group for 8–10 weeks, whereas DBA/2J wild-type mice served as healthy, sedentary controls (WT SED). Selective skeletal muscle mass and *ex vivo* force generation were elevated in D2.mdx animals that ran a relatively high volume (D2.mdx High VWR; $1.84 \pm 0.84$ km/day) compared to low-volume runners (D2.mdx Low VWR; $0.46 \pm 0.31$ km/day) and SED counterparts. VWR did not exacerbate the dystrophy, and instead attenuated the fibrotic profile compared to D2.mdx SED mice. A VWR-induced shift towards a more slow, oxidative phenotype was also observed. Mitochondrial respiration was reduced in D2.mdx SED animals *versus* WT SED mice but was partially restored following both Low and High VWR. Finally, a dose-dependent increase in the expression of mitochondrial proteins was observed following VWR, whereas markers of mitochondrial fusion were particularly elevated in D2.mdx High VWR mice. Our results indicate that VWR enhances muscle and mitochondrial biology in D2.mdx animals and further supports the therapeutic role of exercise for DMD patients.

(Received 16 April 2024; accepted after revision 27 January 2025; first published online 7 April 2025)
**Corresponding author** Vladimir Ljubicic: Department of Kinesiology, McMaster University, Hamilton, ON L8S 4L8, Canada.     Email: ljubicic@mcmaster.ca

**Key points**

- Duchenne muscular dystrophy (DMD) is a life-limiting neuromuscular disorder characterized by muscle weakness and wasting. Skeletal and cardiac muscle quality is compromised in the dystrophic condition.
- Exercise promotes functional and molecular adaptations in healthy individuals and mild dystrophic mouse models. However, the effects of exercise in more severe and clinically relevant models of DMD require investigation.
- A relatively high volume of voluntary wheel running (VWR) augmented selective muscle mass and muscle function without exacerbating the dystrophic pathology in D2.mdx mice.
- Volitional exercise normalized dystrophic skeletal muscle mitochondrial respiration and upregulated mitochondrial content compared to sedentary counterparts. A higher dose of VWR increased organelle fusion protein expression compared to both healthy and dystrophic sedentary animals, as well as D2.mdx mice that ran lower volumes.
- Our results provide evidence from a severe preclinical model that volitional exercise may be a safe and efficacious lifestyle-based intervention for DMD.

**Stephanie Mattina** is a PhD candidate in the Integrative Neuromuscular Biology Laboratory under the supervision of Dr Vladimir Ljubicic. She is a member of the Exercise Metabolism Research Group in the Department of Kinesiology at McMaster University. Her dissertation examines the molecular mechanisms that govern neuromuscular health and disease. Her main research interests include the biology of neuromuscular disorders, ageing and exercise.

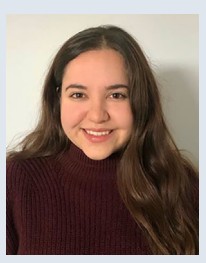

## Introduction

Duchenne muscular dystrophy (DMD) is an X-linked neuromuscular disorder (NMD) that is characterized by progressive and severe muscle weakness and wasting. DMD is the most common form of childhood muscular dystrophy and is caused by mutations in the *DMD* gene that encodes for dystrophin protein (Duan et al., 2021). This results in a loss of functional dystrophin, which mechanically links the actin cytoskeleton to the extracellular matrix. The pathogenesis of DMD is characterized by extensive myofiber degeneration, inflammation and fibrosis across multiple muscle groups, including skeletal, cardiac and respiratory muscles (Duan et al., 2021). Furthermore, evidence from both patients and preclinical models shows reduced mitochondrial quality with altered organelle turnover, dynamics and bioenergetic function (Mikhail et al., 2023). With an incidence of 1 in 3500–5000 male births, boys with DMD present with debilitating symptoms during childhood and ultimately succumb to premature death in their third or fourth decade due to cardiorespiratory complications (Duan et al., 2021; Emery & Eh, 1991; Mendell et al., 2012).

Currently there is no cure for DMD. Glucocorticoids are prescribed as a standard of care to improve muscle function and slow disease progression, despite deleterious side effects after long-term use. Furthermore, recent therapeutic approaches that aim to restore dystrophin expression are limited to specific mutations, and thus, are only applicable to subsets of patients (Roberts et al., 2023). Identifying treatment options that can improve the dystrophic pathology regardless of DMD mutations, or show additive benefits when combined with gene therapies, would aid in improving the lives of individuals with DMD. Prescribed exercise is a safe, efficacious and cost-effective lifestyle intervention that can promote functional and molecular adaptations in healthy individuals, as well as in those with various NMDs (Furrer et al., 2023; Mikhail et al., 2022; Ng et al., 2018). Aerobic exercise improves muscle health and function in both DMD patients and animal models, despite concern due to increased vulnerability of dystrophic muscle (Spaulding & Selsby, 2018). It is generally accepted that high-intensity exercise and downhill running with eccentric contractions (ECCs) exacerbate the dystrophic pathology, whereas lower-intensity physical activity leads to beneficial adaptations in the absence of muscle damage (Hyzewicz et al., 2015). These findings have primarily been documented in C57.mdx mice; however this commonly studied mouse strain displays a mild pathology, so additional investigations that use more severe and clinically relevant models of DMD, such as D2.mdx animals, are necessary (Swiderski & Lynch, 2021). Recent literature that examined the effects of endurance-type treadmill run training in D2.mdx mice demonstrates improved force production and reduced markers of the dystrophic pathology (Zelikovich et al., 2019). In contrast to these forced activity modalities, rodent voluntary wheel running (VWR) is a form of self-regulated exercise, which elicits a myriad of positive morphological and functional adaptations, including in C57.mdx mice (Hamm et al., 2021; Hyzewicz et al., 2015). Further investigation of volitional exercise in DMD could inform future preclinical studies that aim to uncover therapeutic avenues that modulate the dystrophic pathology. Thus the purpose of this study was to investigate the physiological and molecular effects of VWR in the more severe, preclinical D2.mdx model of DMD. We hypothesized that volitional exercise would promote multisystemic adaptations, without exacerbating the dystrophic phenotype.

## Methods

### Animals

Seven-to-eight-week-old healthy DBA/2J wild-type (WT) and dystrophic D2.mdx mice were obtained from Jackson Laboratory (WT: 000671, D2.mdx: 013141) and used in this study. The D2.mdx animals were engineered on the DBA background that carry a polymorphism in the latent transforming growth factor $\beta$ (TGF-$\beta$) binding protein 4 gene (*Ltbp4*), which is also a genetic modifier of disease severity in DMD patients (Flanigan et al., 2013; Mázala et al., 2020). Although the D2.mdx model does not fully recapitulate the extent of degeneration and fatty tissue accumulation seen in DMD patients and exhibits distinct intramuscular calcification, it remains a more accurate phenocopy compared to the typical C57.mdx mouse model (Coley et al., 2016; Hammers et al., 2020; Swiderski & Lynch, 2021). Animals were housed under a 12 h light/dark cycle in environmentally controlled conditions and were provided food and water *ad libitum*.

### Volitional exercise protocol and tissue collection

In this study, 16 male D2.mdx mice were assigned to a sedentary (D2.mdx SED) group, 27 male D2.mdx mice to a VWR (D2.mdx VWR) condition and 16 male WT mice served as healthy, sedentary controls (WT SED). Only male mice were used in this study because DMD is an X-linked disorder that primarily affects only boys and young men. This study did not explore the effects of VWR in WT mice, as extensive research already exists on the effects of exercise in unaffected, healthy mice (Smith et al., 2023). D2.mdx VWR animals were individually housed with access to a home-cage running wheel (Columbus Instruments, Columbus, OH, USA) for 8–10 weeks. The number of revolutions was automatically

recorded every 10 min throughout the intervention using an Opto-M3 activity metre with the Multi-Device Interface software (version 1.5, Columbus Instruments). The distance travelled was determined using the number of revolutions recorded by the tracking software and circumference of the running wheel. Human observations were made throughout the intervention to confirm that the tracked revolutions reflected true running behaviour with no external manipulation. At the end of the intervention, exercise wheels were removed from the cages, and after 12 h of rest, *in vivo* functional tests were completed, and body composition was measured. Following a 48 h washout period without access to the running wheels, mice were weighed, and *ex vivo* functional tests were conducted. WT and D2.mdx mice were killed via cervical dislocation, and muscle tissues were collected. The tibialis anterior (TA), extensor digitorum longus (EDL), gastrocnemius (GAST), soleus (SOL), quadriceps (QUAD), triceps (TRI) and diaphragm (DIA) muscles were harvested and immediately snap frozen in liquid nitrogen. Contralateral skeletal muscles, as well as the heart, were weighed, mounted in optimal cutting temperature (OCT; Thermo Fisher Scientific, Waltham, MA, USA), frozen in isopentane and cooled in liquid nitrogen. All tissues were stored at −80°C until further analysis.

## Body composition measurement

After the intervention, percent fat and lean mass were assessed using the Minispec Whole Body Composition Analyzer (Bruker, Billerica, MA, USA). Body weight was measured prior to the experiment, and percent fat and lean mass were calculated as grams of fat or lean tissue/g body weight × 100.

## Skeletal muscle functional testing

D2.mdx and WT animals underwent two rounds of familiarization testing for each *in vivo* functional experiment during the last week of the VWR protocol, including forelimb grip strength and open-field activity trials. After the exercise intervention, functional tests were conducted. Forelimb grip strength was measured in accordance with the Treat-NMD SOP: DMD_M.2.2.001. Briefly, the animals grasped on a wire grip dynamometer (Columbus Instruments) with their forelimbs, and force transduction was measured as the mice were pulled away until their grasp was lost. Mice were then returned to their cage for 5 min of rest, and 5 × 3 trials were conducted. The average of the trial with the highest force was taken as the maximum grip strength measure and normalized to body weight in grams. Activity levels and locomotor behaviour were measured using the Opto-Varimex-5 Auto-Track

System (Columbus Instruments) and modified from the Treat-NMD SOP: DMD_M.2.1.002. Prior to data collection, animals were acclimatized in the arena for 30 min to ensure familiarization with the testing environment. On the following day, animals were placed in the activity chambers and data were recorded for 1 h in an undisturbed environment. Several activity metrics were quantified, including ambulatory time and rearing events.

Functional testing was performed *ex vivo* using the whole-muscle test system (Aurora Scientific, Aurora, ON, Canada), as described earlier (Wang et al., 2023). Briefly, after the VWR intervention, the EDL muscle was isolated, and the proximal and distal tendons were fixed to a stationary lever and force transducer (Aurora Scientific) at an optimal length using braided silk suture. The EDL was placed in oxygenated Ringer solution (120 mM NaCl, 4.7 mM KCl, 2.5 mM CaCl2, 1.2 mM KH2PO4, 1.2 mM MgSO4, 25 mM HEPES, 5.5 mM glucose) and submerged for 10 min. The muscle was stimulated for 1 s every 30 s at increasing stimulation frequencies, starting at 10 Hz and increasing in 10 Hz increments up to 140 Hz, to generate a force–frequency curve. Next, an ECC protocol was performed to measure fatigue. In brief, 10 ECCs were initiated with a 700 ms train duration of supra-maximal 10 V, 0.2 ms square pulses at 200 Hz, with a 10% lengthening at a velocity of 0.5 Le/s during the last 200 ms, conducted at 2 min intervals. Data were collected and analysed using the Dynamic Muscle Control and Analysis Software (Aurora Scientific), and values were normalized to EDL muscle mass.

## Preparation of permeabilized fibres and mitochondrial respiration

Myofiber permeabilization and mitochondrial oxygen consumption were measured using high-resolution respirometry techniques with the Oxygraph-2k (Oroboros Instruments, Innsbruck, Austria), as previously described (Hughes et al., 2019; Stouth et al., 2023). In short, during tissue collection, the QUAD muscle was rapidly excised, and a section was placed in ice-cold biopsy preservation solution (BIOPS) buffer (50 mM K-MES, 7.23 mM K2EGTA, 2.77 mM CaK2EGTA, 20 mM imidazole, 20 mM taurine, 5.7 mM ATP, 14.3 mM phosphocreatine and 6.56 mM $MgCl_2$, pH 7.1). Connective tissue and fat were trimmed from myofibers, and fine-tip forceps were used to further separate myofibers into muscle bundles under a dissection microscope on a liquid nitrogen cooled block. During permeabilization, fibre bundles were placed in saponin (50 μg/ml) in BIOPS and rolled end-over-end for 30 min at 4°C. Finally, myofiber bundles were placed in MIR05 (0.5 mM EGTA, 10 KH2PO4, 3 mM MgCl2•6 H2O, 60 mM K-lactobionate, 20 mM

HEPES, 20 mᴍ taurine, 110 mᴍ sucrose and 1 g/l BSA; pH 7.1) at 4°C for 15 min prior to oxygen consumption measurements.

Oxygen calibration was performed on the Oxygraph-2k chambers prior to the experiment. Respirometry measurements were conducted in 2 ml of MIR05 buffer containing blebbistatin (5 µM). After myofiber permeabilization, muscle bundles were placed in the chambers at 37°C. First, pyruvate (5 mᴍ) and malate (2 mᴍ) were introduced to stimulate complex I (CI), followed by the addition of maximal ADP (5 mᴍ). Once steady state was reached, glutamate (5 mᴍ) and succinate (20 mᴍ) were added to determine CI- and CI+II-supported respiration, respectively. Cytochrome $c$ (10 mᴍ) was added to ensure mitochondrial membrane integrity. Respiration values were normalized to respective muscle bundle wet weights.

## Protein extraction and quantification

A fraction of frozen GAST muscles (∼30 mg) were crushed in a tissue pulverizer (CellCrusher, Portland, OR, USA). The apparatus was first submerged in a liquid nitrogen bath, and muscle tissue was removed from liquid nitrogen and placed in the mortar component. The pestle was positioned on top, and a mallet was used to homogenize tissue. Powdered muscle was transferred to a 1.5 ml tube with radioimmunoprecipitation assay (RIPA) buffer (20 µL of RIPA buffer per 1 mg muscle weight; Sigma Aldrich) supplemented with protease and phosphatase inhibitor cocktail tablets (Roche, Laval, QC, Canada). The samples were then sonicated (Thermo Fisher Scientific) on ice for 5 × 5 s at 70% power and were rolled end-over-end for 30 min at 4°C. After this the samples were centrifuged at 14,000 $g$ for 10 min at 4°C, and supernatants were collected. Protein concentration was measured using a bicinchoninic assay (BCA; Thermo Fisher Scientific). Samples were diluted to a final concentration of 2 µg/µl using ddH$_2$O and equal amounts of 4× loading buffer.

## Western blotting

To investigate muscle protein expression, 20 µg of each sample was loaded into the lanes of 4%–15% Criterion TGX gels (Bio-Rad Laboratories, Mississauga, ON, Canada) and separated using SDS-PAGE at 190 V for 1 h. Stain-free gels were activated, and proteins were transferred onto nitrocellulose membranes with the Trans-Blot Turbo System (Bio-Rad Laboratories). Next, membranes were imaged using the ChemiDoc MP Imaging System (Bio-Rad Laboratories) to normalize total protein content. Membranes were then washed in 1× Tris-buffered saline with 0.1% Tween 20 (TBST) for 3 × 5 min and placed in a blocking solution containing 5% bovine serum albumin (BSA) in TBST for 1 h at room temperature. Subsequently, membranes were incubated with primary antibodies listed in Table 1 overnight at 4°C. On the following day blots were then washed in 1× TBST for 3 × 5 min and placed in the appropriate secondary antibody for 1 h at room temperature. Next, membranes were washed in 1× TBST for 3 × 5 min, and target proteins were detected using enhanced chemiluminescence substrate (Bio-Rad Laboratories). Images were taken with the ChemiDoc MP Imaging System, and densitometry was analysed using Image Lab software (Bio-Rad Laboratories). All blots were normalized to their respective stain-free membrane and standardized to a consistent loading control on each gel.

## Histological analyses

Haematoxylin and eosin (H&E) staining was performed on SOL and DIA muscles to quantify the percentage of centrally nucleated fibres (CNF). Muscles were sectioned at 10 µm using a cryostat (Thermo Fisher Scientific) at −20°C. Muscle cross-sections were stained with H&E (Sigma-Aldrich, St. Louis, MO, USA), followed by dehydration with 95% and 100% ethanol, further dried with xylene and mounted with Permount medium (BioWorld, Dublin, ON, USA). CNFs were quantified on entire muscle cross-sections by counting the number of total myofibers, as well as the number of myofibers containing one or more centrally located nuclei. A Masson trichrome staining kit (Sigma-Aldrich) was used to measure fibrosis, as indicated by the accumulation of collagen in the GAST, DIA and heart muscles. Briefly, tissues were sectioned at 10 µm and fixed with 4% paraformaldehyde (PFA) for 1 h followed by Bouin's fixative overnight at room temperature. The following day tissues were washed and placed in working Weigert's iron hemotoxylin solution (EMD Millipore, Billerica, MA, USA) for 5 min. Cross-sections were then placed in running wash tap water for 10 min and incubated in Biebrich scarlet-acid fuchsin for 15 min. Samples were washed, placed in working phosphotunstic and phosphomolydic acid solution for 3 × 3 min, and incubated in aniline blue for 5 min. Tissues were washed, placed in 1% glacial acetic acid, washed, dehydrated with 95% and 100% ethanol, xylene, and mounted with Permount. Whole muscles were analysed for percent collagen-stained area using a semi-automated method. Histological stains were imaged using light microscopy at 20× magnification on Nikon Elements Microscopic Imaging Software (Nikon Instruments, Mississauga, ON, Canada) and analysed using NIS Elements. Investigators were blinded to each group during the analysis process for all histological stains.

**Table 1. List of primary antibodies.**

Western blotting

| Protein | Manufacture | Product # | Host | Primary dilution | Secondary dilution |
|---|---|---|---|---|---|
| Citrate synthase | Abcam | ab96600 | Rabbit | 1:1,000 | 1:10, 000 |
| DRP1 | CST | 8570 | Rabbit | 1:1,000 | 1:10, 000 |
| p-DRP1[Ser637] | CST | 4867 | Rabbit | 1:1,000 | 1:10, 000 |
| FIS1 | Proteintech | 10 956-1-AP | Rabbit | 1:1,000 | 1:10, 000 |
| OPA1 | Abcam | ab42364 | Rabbit | 1:1,000 | 1:10, 000 |
| OXPHOS | Abcam | ab110413 | Mouse | 1:1,000 | 1:10, 000 |
| MFN2 | Abcam | ab56889 | Mouse | 1:1,000 | 1:10, 000 |
| PGC-1$\alpha$ | EMD Millipore | ab3242 | Rabbit | 1:1,000 | 1:10, 000 |
| SOD2 | Abcam | ab11575 | Rabbit | 1:1,000 | 1:10, 000 |

Abbreviation: CST, cell-signalling technologies.

## Immunofluorescence and confocal microscopy

GAST and SOL muscles were cut at 10 µm on a cryostat at −20°C. GAST muscles were stained for IgG as a marker of membrane integrity. In brief, samples were incubated overnight with goat anti-mouse IgG Alexa Fluor 488 (1:500; A11029; Thermo Fisher Scientific), washed 3 × 5 min with 1× phosphate-buffered saline (PBS) and mounted with Prolong Gold antifade reagent (Thermo Fisher Scientific). SOL muscles were stained for myosin heavy chain (MHC) isoforms, as previously described (Bloemberg & Quadrilatero, 2012). Tissues were placed in 10% goat serum in PBS for 1 h, followed by application of primary antibody cocktail against MHC I (1:50; BA-F8), MHC IIa (1:600; SC-71), MHC II× (1:50; 6H1; Developmental Studies Hybridoma Bank, Iowa City, IA, USA) and Laminin (1:500; L0663; Sigma-Aldrich) for 2 h. Slides were washed 3 × 5 min with 1× PBS, followed by incubation with the appropriate isotype-specific fluorescent secondary antibodies (1:500; Invitrogen, Carlsbad, CA, USA). Whole-muscle cross-sections were imaged with the Nikon Elements Microscopic Imaging Software at 20× magnification (Nikon Instruments) and analysed on NIS Elements. Regions of the GAST muscle containing IgG infiltration were manually highlighted using a binary threshold, and a percent was calculated within the entire muscle. In the MHC stain, a binary threshold was generated with the laminin staining and confirmed manually by a blinded investigator. Next, a binary threshold was generated for MHC I, IIa and IIx, which was subtracted from laminin to obtain a cleaner binary threshold. Fibre type minimum (min) feret diameter per fibre type was confirmed, and hybrid fibres were manually placed in a separate binary layer. Fibre type distribution and min feret diameter were determined using the automated measurement results tool in NIS Elements. All investigators were blinded to all samples prior to analysis.

Rapid dissection of the epitrochleoanconeus (ETA) muscle and wholemount immunohistochemical labelling of pre- and postsynaptic components were performed as previously described (vanLieshout et al., 2022; Villarroel-Campos et al., 2022). The ETA muscle was pinned to a Slygard-coated Petri dish and isolated under a dissection microscope. Next, muscles were fixed for 10 min in 4% PFA at room temperature and permeabilized in cold methanol at −20°C for 6 min. Muscles were incubated with 10% goat serum in PBS containing 0.1% Triton X-100 for 1 h at room temperature to prevent non-specific binding. Next, samples were incubated overnight for motor axons with mouse IgG1 anti-neurofilament M (1:50; 2H3; Developmental Studies Hybridoma Bank) and nerve terminals with mouse IgG1 anti-synaptic vesicular protein 2 (1:100; SV2; Developmental Studies Hybridoma Bank) at 4°C. On the next day, muscles were incubated with goat anti-mouse IgG Alexa-594 secondary antibody (1:500; Thermo Fisher Scientific) for 1 h, followed by postsynaptic acetylcholine receptor (AChR) labelling with Alexa-488 conjugated-$\alpha$-bungarotoxin (1:500; Invitrogen) for 1 h. Samples were washed 3 × 5 min in PBS containing 0.01% Triton X-100 between each step. Tissues were mounted using Prolong Gold antifade reagent (Thermo Fisher Scientific), and images were captured using confocal microscopy on a 60× oil objective (Nikon Instruments). Twenty-to-forty neuromuscular junctions (NMJs) were imaged per sample, and the number of fragmented AChR clusters per NMJ was manually counted in a Z-stack using NIS Elements.

## Statistical analyses

The data presented in this study aim to investigate the effect of volitional exercise on the dystrophic pathology in D2.mdx animals. Data normality was assessed using the Shapiro–Wilk test. Normally distributed data were

analysed using parametric one-way ANOVA tests to compare group means. Furthermore, two-way ANOVA tests were conducted to assess main effects of time, frequency or contraction and group, where appropriate. Correlations were analysed using simple linear regression. Pearson correlation coefficient (r) effect sizes were classified as small (r = 0.1), medium (r = 0.3) or large (r = 0.5). If significance was reached, comparisons between groups were determined using a Tukey's *post hoc* test. The non-parametric Kruskal–Wallis test was used for non-normally distributed data, and multiple comparisons were assessed using a Dunn's *post hoc* test. *p*-Values < 0.05 were considered statistically significant. All statistical analyses were performed using the GraphPad Prism 10 software. Data are presented as mean ± SD.

## Results

### Voluntary running augments muscle mass and function in D2.mdx mice

We first aimed to characterize the voluntary running behaviour in the D2.mdx model of DMD. Seven-to-eight-week-old male D2.mdx mice were provided access to a home-cage running wheel for 8–10 weeks (Fig. 1*A*). The average running distance of the D2.mdx animals throughout the intervention was 1.18 ± 0.62 km/day (Fig. 1*B*). As expected, this was lower than the average daily running volumes in both C57.mdx and WT animals (Hayes & Williams, 1996; Lerman et al., 2002; Turner et al., 2005). Consistent with previous literature in C57.mdx animals (Kogelman et al., 2018; Smythe & White, 2012),

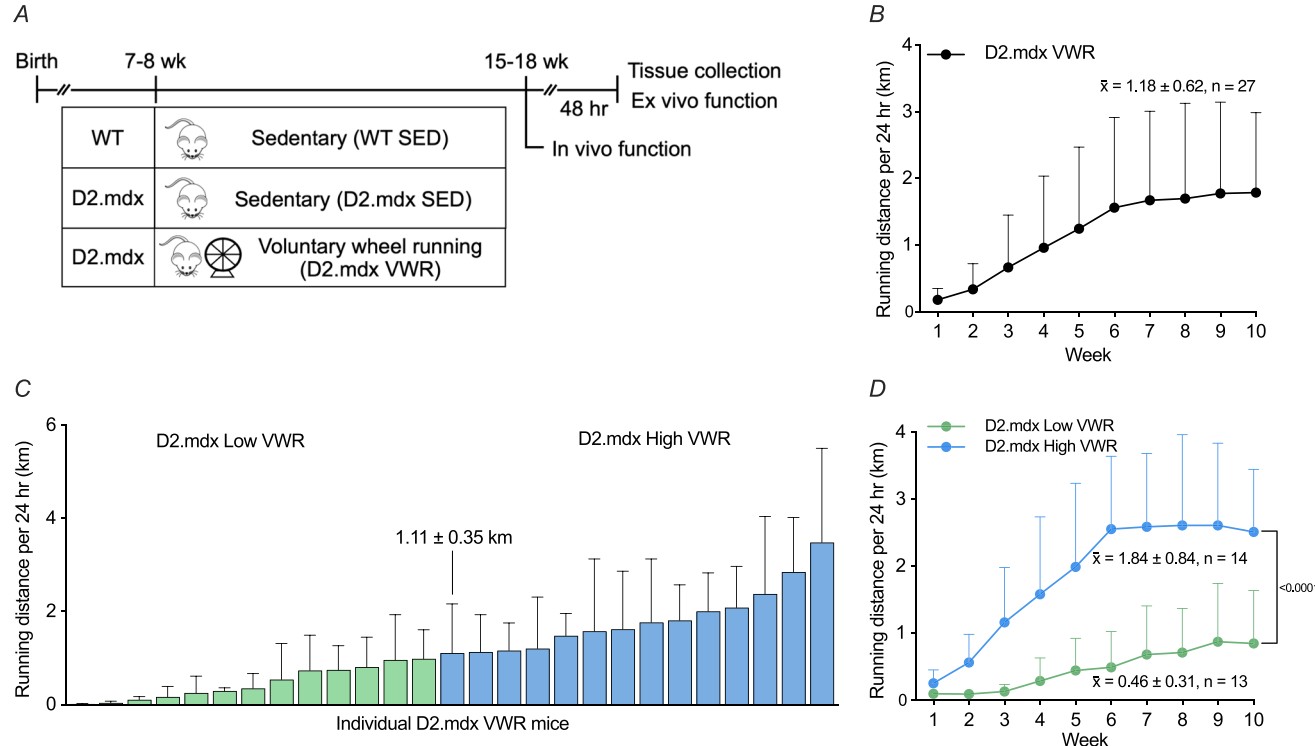

**Figure 1.  Weekly volitional exercise in D2.mdx mice**
*A*, schematic overview of the experimental design. Seven to eight-week-old male mice were monitored for 8–10 weeks (wk). DBA/2J wild-type mice served as sedentary controls (WT SED), whereas D2.mdx mice were randomly assigned to a sedentary (D2.mdx SED) or a voluntary wheel running (D2.mdx VWR) group. Mice randomized to the VWR group were given access to a home-cage running wheel for the duration of the intervention. *In vivo* functional tests were performed following the intervention at 15–18 weeks of age. Tissues were collected, and *ex vivo* muscle function was measured 48 h (hr) postintervention. *B*, average daily running distance per week (in kilometres) in the D2.mdx VWR group. *C*, average daily running distance of each D2.mdx VWR mouse throughout the intervention. The D2.mdx VWR group was separated using the median running distance (1.11 ± 0.35 km) to compare mice that ran a relatively low (D2.mdx Low VWR) and high (D2.mdx High VWR) volume. *D*, average daily running distance per week (in kilometres) in the D2.mdx Low and High VWR groups. D2.mdx VWR: *n* = 27 (D2.mdx Low VWR: *n* = 13; D2.mdx High VWR: *n* = 14). Data are presented as mean ± SD. For (*D*), a two-way ANOVA was used to assess the main effects of time and group.

VWR activity in the D2.mdx mice was variable, ranging from 0.02–3.48 km/day (Fig. 1*C*). The D2.mdx VWR mice were partitioned into groups that ran a relatively low (D2.mdx Low VWR) and high (D2.mdx High VWR) volume using the median daily running distance (i.e. 1.11 ± 0.35 km/day). The D2.mdx Low VWR animals ran significantly less than the D2.mdx High VWR group, running an average of 0.46 ± 0.31 km/day *versus* 1.84 ± 0.84 km/day, respectively (Fig. 1*D*).

Next, we assessed body and skeletal muscle mass in response to VWR in D2.mdx animals. Body mass was significantly reduced in D2.mdx SED animals compared to WT SED controls, with no difference between D2.mdx VWR mice compared to either WT or D2.mdx SED animals (van Putten et al., 2019) (Fig. 2*A*). Furthermore, fat and lean mass remained similar between WT SED, D2.mdx SED, D2.mdx Low VWR and D2.mdx High VWR mice (Fig. 2*B*). Relative heart mass was significantly elevated in D2.mdx High VWR mice *versus* WT SED mice, with no difference between WT SED, D2.mdx SED and D2.mdx Low VWR animals (Fig. 2*C*). D2.mdx mice showed a significant reduction in normalized muscle mass of the TRI, QUAD and GAST muscles compared to WT SED animals (Fig. 2*D*). However, normalized mass was significantly higher (+ 13%–15%) in D2.mdx mice that ran a relatively high volume compared to their SED counterparts in the TRI ($p = 0.0596$), QUAD and GAST muscles, as well as in the D2.mdx Low VWR group in the QUAD. Normalized mass of the TA was significantly reduced in D2.mdx Low and High VWR mice compared to WT SED animals, whereas the EDL muscle remained unchanged across all four experimental groups (Fig. 2*E*). Of note, the normalized mass of the SOL, a plantar flexor muscle that is heavily recruited while running, was significantly higher in the D2.mdx High VWR (+ 55%) and D2.mdx Low VWR (+ 20%) mice compared to the D2.mdx SED group. A genotype difference was also observed, with significantly increased relative SOL mass in D2.mdx animals *versus* WT SED controls.

To determine the effects of voluntary exercise on skeletal muscle function in dystrophic mice, we performed a series of *in vivo* and *ex vivo* physiological tests. Relative forelimb grip strength was significantly reduced in D2.mdx animals compared to healthy WT SED mice, with no difference between D2.mdx animals (Fig. 2*F*). Metrics of locomotor behaviour during a 1 h activity index, including ambulatory time and rearing events, were significantly lower (−36%–40%) in D2.mdx SED and Low VWR mice compared to WT SED animals (Fig. 2*G* and *H*). Ambulatory time and rearing events in D2.mdx mice that ran a relatively higher volume were not statistically different from (−17%–26%) WT SED or D2.mdx (+22%–30%) animals. To further characterize skeletal muscle function following chronic volitional exercise, we assessed *ex vivo* force generation of the EDL muscle. The muscle was isolated from D2.mdx animals and stimulated at frequencies ranging from 0 to 140 Hz. Force generation was significantly greater in D2.mdx mice that ran a relatively higher volume compared to D2.mdx Low VWR (−26%) and D2.mdx SED (−31%) animals (Fig. 2*I*). It is well known that dystrophic muscle is highly susceptible to ECC-induced damage (Head et al., 1992; Petrof et al., 1993). Therefore we subjected EDL muscles to a train of 10 ECCs and measured force production. In line with previous work (Hogarth et al., 2017; Olthoff et al., 2018), we observed a decline in tension following each ECC in dystrophic muscle (Fig. 2*J*). A strong statistical trend was detected ($p = 0.0535$) with force production during ECCs, where mice that ran a relatively higher volume of volitional exercise generated 23%–26% more force than D2.mdx SED and D2.mdx Low VWR mice. Notably, the preservation of force production, expressed as a percentage of the first ECC, was significantly greater in D2.mdx High VWR animals compared to both D2.mdx SED and D2.mdx Low VWR mice (Fig. 2*K*).

## Favourable exercise-related adaptations in skeletal and respiratory muscle histology and phenotype

We investigated the impact of our volitional exercise intervention on skeletal, respiratory and cardiac muscle quality in D2.mdx mice. To characterize the influence of volitional exercise on the degenerative myopathy in dystrophic muscle, we measured the presence of CNFs. We observed a similar level of CNFs in the SOL and DIA muscles between D2.mdx groups, with significantly higher CNFs compared to WT mice (Fig. 3*A* and *B*). IgG infiltration in the GAST muscle as a marker of skeletal muscle integrity was significantly increased in D2.mdx SED and High VWR animals compared to healthy controls, with no statistical difference between D2.mdx Low VWR and WT SED animals (Fig. 3*C* and *D*). Additionally, collagen deposition as a measure of fibrosis was quantified in the GAST, heart and DIA muscles (Fig. 3*E–H*). Fibrosis was significantly higher in the GAST and DIA muscles of D2.mdx groups compared to WT SED animals. Notably, fibrotic area was significantly lower in D2.mdx High VWR mice compared to their SED counterparts. In the GAST muscles, the D2.mdx Low VWR group also exhibited significantly less fibrosis compared to D2.mdx SED animals. No difference in collagen deposition was observed in the heart across the four experimental groups.

The structure and function of the NMJ are altered in both DMD patients and preclinical C57.mdx mice (Ng & Ljubicic, 2020). Thus we analysed the morphology of the postsynapse within the ETA, which is a supinator muscle located in the upper forelimb that is commonly used to assess NMJ biology, because its thin structure is

highly amenable to synaptic imaging (Villarroel-Campos et al., 2022). The number of AChR clusters, defined by discontinuous fragments of the postsynaptic endplate, was significantly higher in D2.mdx animals compared to their SED WT counterparts, whereas D2.mdx SED mice remained similar to D2.mdx Low VWR ($p = 0.0695$) and D2.mdx High VWR ($p = 0.0852$) groups (Fig. 4*A–C*).

We then examined the influence of chronic exercise on myofiber characteristics in dystrophic muscle. We first assessed exercise adaptations in fibre type composition

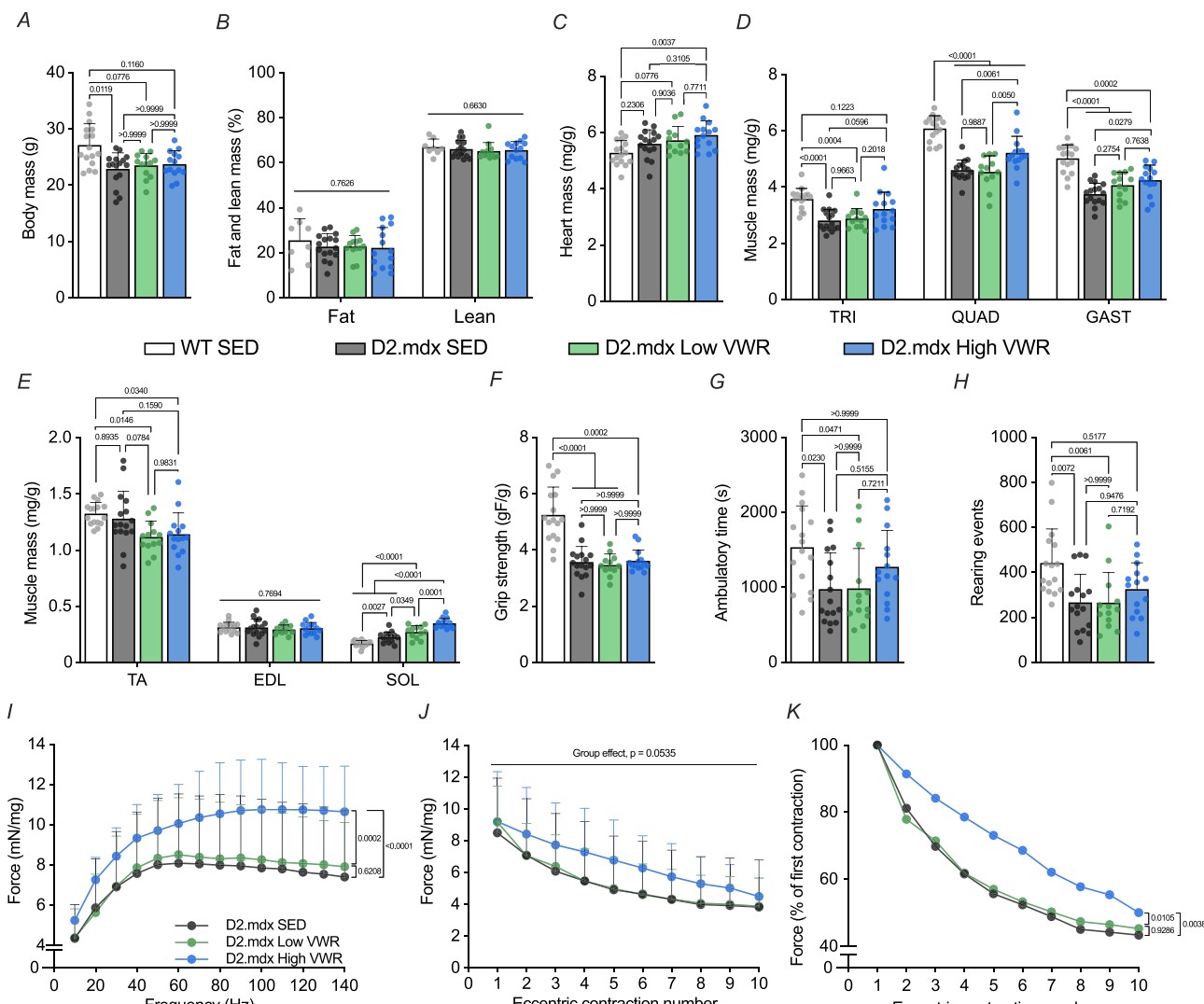

**Figure 2. A relatively high volume of volitional exercise augments selective muscle mass and specific muscle function in D2.mdx mice**

*A*, body mass (in g) of WT SED, D2.mdx SED, D2.mdx Low VWR and D2.mdx High VWR groups postintervention. *B*, body composition, including the percentage of fat and lean mass relative to body mass. *C*, heart mass expressed relative to body mass (in mg/g). *D* and *E*, mass of the triceps (TRI), quadriceps (QUAD), gastrocnemius (GAST), tibialis anterior (TA), extensor digitorum longus (EDL) and soleus (SOL) muscles expressed relative to body mass. *F*, maximum forelimb grip strength expressed relative to body mass (gram force per gram; gF/g) following the intervention. *G* and *H*, ambulatory time (s) and rearing events during a 1 h open-field activity index postintervention. *I*, force–frequency relationship showing the specific EDL force generation (millinewton per milligram EDL mass; mN/mg) at increasing frequencies from 0 to 140 Hz in D2.mdx animals. *J*, specific EDL force production during 10 eccentric contractions (ECCs) postintervention. *K*, force drop with each ECC expressed as a percentage of initial force. *A–H*, $n = 13–16$; *I–K*, $n = 3–14$. Data are presented as mean ± SD. For (*A–H*), a one-way ANOVA or the Kruskal–Wallis test was used, followed by a Tukey's or Dunn's multiple comparisons test where appropriate. For (*I*) and (*K*), a two-way ANOVA followed by a Tukey's multiple comparisons test was used to assess the main effects of frequency or contraction and group. For (*J*), a two-way ANOVA was used to assess the main effects of contraction and group.

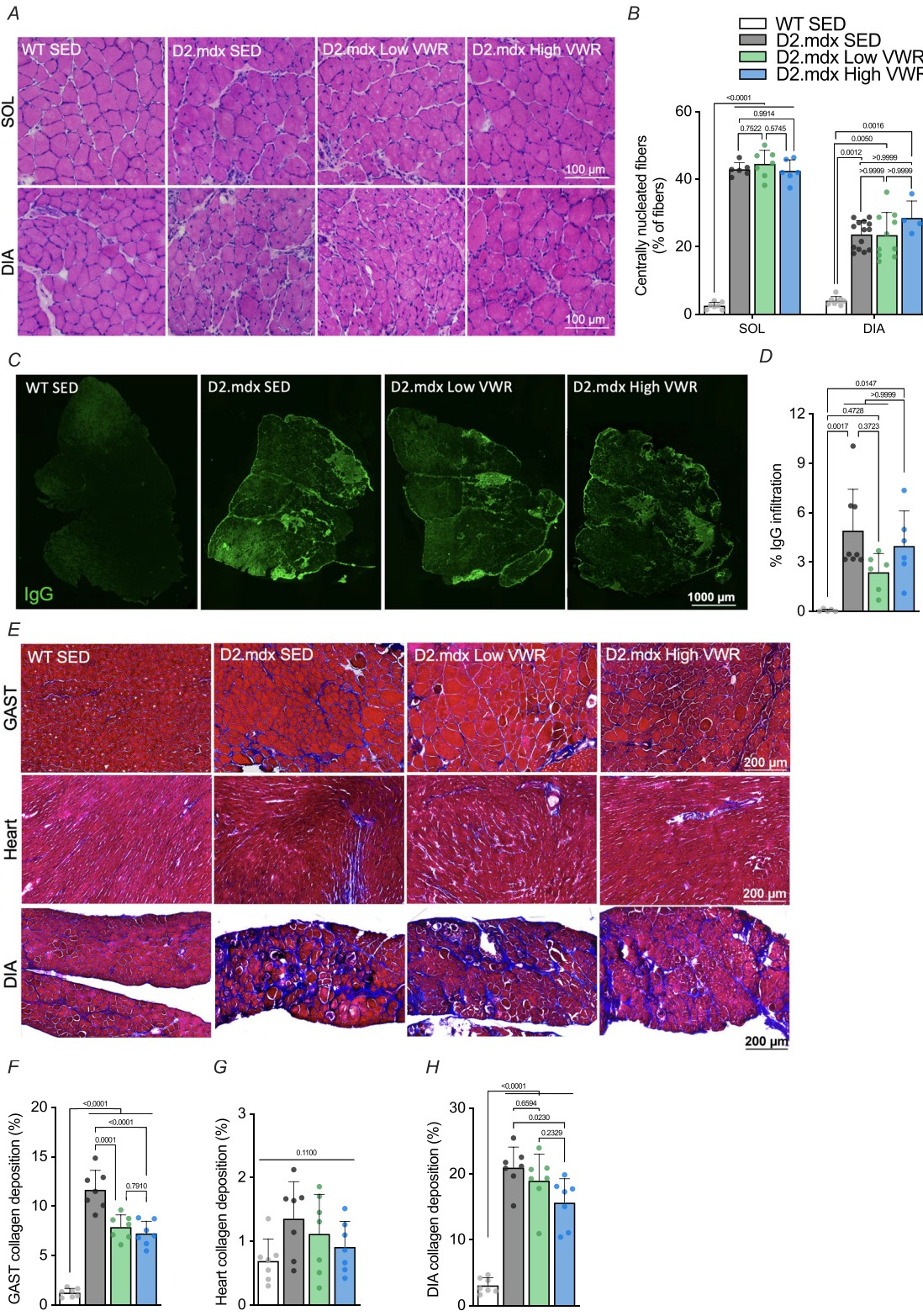

**Figure 3. Volitional exercise improves the dystrophic phenotype in D2.mdx animals**

*A*, representative haematoxylin and eosin (H&E) stains of soleus (SOL) (top panel) and diaphragm (DIA, bottom panel) muscle cross-sections. Scale bar = 100 μm. *B*, quantification of centrally nucleated fibres in the SOL and DIA muscles across the four experimental groups. *C*, typical immunofluorescence (IF) images of IgG in gastro-cnemius (GAST) muscles as a measure of muscle membrane integrity following the intervention. Scale bar = 1000 μm. *D*, graphical summary of the percentage IgG infiltration within the muscle in WT and D2.mdx animals. *E*,

representative Masson trichrome stains of the GAST (top panel), heart (middle panel) and DIA (bottom panel) across the four experimental groups. Scale bar = 200 μm. Quantification of the percentage collagen deposition within cross-sections of the (*F*) GAST, (*G*) heart and (*H*) DIA muscles. *A–D*, n = 5–14; *E–H*, n = 7. Data are presented as mean ± SD. For (*B*, *D*, *F–H*), a one-way ANOVA or the Kruskal–Wallis test was used, followed by a Tukey's or Dunn's multiple comparisons test where appropriate.

and min feret diameter in D2.mdx mice. We observed a statistical trend ($p = 0.0556$) in the percentage of type I fibres (Fig. 4*D* and *E*). Specifically, on average, there were 11%–14% more type I fibres in D2.mdx mice that exercised compared to both WT SED and D2.mdx SED controls. The percentage of MHC type IIa and hybrid fibres remained similar between WT and D2.mdx animals. Other fibre types, which include type IIb, type IIx or embryonic MHC-positive fibres, were significantly reduced in D2.mdx mice that ran a relatively high volume compared to both WT and D2.mdx SED. Additionally, the variance coefficient of min feret diameter, which is a measure of fibre size variability, was similar between D2.mdx groups but significantly higher than WT SED animals (Fig. 4*D* and *F*). Consistently, min feret diameter was also significantly different in WT SED *versus* D2.mdx animals in type I, type IIa and cumulative fibre types (Fig. 4*D* and *G*). The percentage of small fibres with a min feret dimeter of ∼10 μm in size was significantly higher in D2.mdx SED animals compared to D2.mdx High VWR mice, which was largely attributed to type I fibres ($p = 0.0969$).

## A relatively high volume of volitional exercise is accompanied by increased mitochondrial respiration, content and fusion in D2.mdx mice

Chronic aerobic exercise is a robust stimulus to induce mitochondrial adaptations within skeletal muscle (Furrer et al., 2023). However, impaired skeletal muscle mitochondrial biogenesis and function have been observed in both DMD patients and animal models (Mikhail et al., 2023). We sought to investigate the mitochondrial adaptations in response to long-term volitional exercise by measuring organelle respiratory activity in the QUAD muscle. We found that CI-supported state 2 respiration was similar between both WT and D2.mdx animals (Fig. 5*A*). D2.mdx SED animals exhibited significantly reduced CI state 3 respiration compared to WT SED (− 59%) and D2.mdx High VWR (− 57%) mice (Fig. 5*B*). Furthermore, CI+II-supported state 3 respiration was significantly lower (− 40%–49%) in D2.mdx SED mice compared to healthy WT counterparts, D2.mdx Low and D2.mdx High VWR animals, which remained similar (Fig. 5*C*).

To complement the respirometry experiments, we next assessed mitochondrial protein content. In the GAST muscles, the expression of PGC-1α was similar between WT SED, D2.mdx SED and D2.mdx Low VWR animals, whereas D2.mdx High VWR animals were significantly (∼1.2–1.3-fold) greater than WT and D2.mdx SED groups (Fig. 5*D* and *E*). Daily running distance in D2.mdx animals was significantly correlated (r = 0.61) with skeletal muscle PGC-1α content (Fig. 5*F*). The expression of mitochondrial oxidative phosphorylation (OXPHOS) protein content in CI-V remained similar between WT SED and D2.mdx SED groups. We observed a significant exercise-induced increase in OXPHOS protein expression in D2.mdx Low and High VWR groups *versus* WT SED mice (Fig. 5*D* and *G*). Furthermore, the expression of CI, CIV and CV was significantly higher in D2.mdx mice that ran a relatively high volume compared to their D2.mdx SED counterparts. The protein expression of CIII was significantly increased in the D2.mdx mice that ran a low relative volume compared to D2.mdx SED mice. Cumulative protein content of OXPHOS complexes was significantly higher in D2.mdx mice that exercised compared to both D2.mdx SED and WT SED animals. Furthermore, we observed a significantly increased expression of citrate synthase (CS) content in D2.mdx Low and High VWR mice compared to WT SED animals, whereas its expression was comparable between D2.mdx groups (Fig. 5*D* and *H*). The expression of superoxide dismutase 2 (SOD2) remained similar in WT SED, D2.mdx SED and D2.mdx Low VWR groups. The increased mitochondrial protein content following a relatively high volume of exercise was paralleled by a significant increase in SOD2 protein expression in D2.mdx High VWR mice compared to their SED counterparts (Fig. 5*D* and *I*).

Finally, we explored the effects of exercise on mitochondrial dynamics in the dystrophic condition. In GAST muscles, total dynamin-related protein 1 (DRP1) expression was significantly higher in D2.mdx SED and D2.mdx High VWR mice compared to healthy controls (Fig. 5*D* and *J*). Inhibitory phosphorylation at DRP1[Ser637] was significantly (∼1.5–1.8-fold) greater in the D2.mdx High VWR mice compared to WT SED, D2.mdx SED and D2.mdx Low VWR groups, which remained similar. DRP1[Ser637] phosphorylation status (i.e. the phosphorylated form of the protein relative to its total content) was significantly (∼1.5-fold) greater in D2.mdx mice that ran a relatively high volume compared to the remaining D2.mdx groups. Despite this, DRP1[Ser637] phosphorylation status in WT SED animals was not statistically different from D2.mdx SED, Low VWR or High VWR groups. A genotype effect was observed with

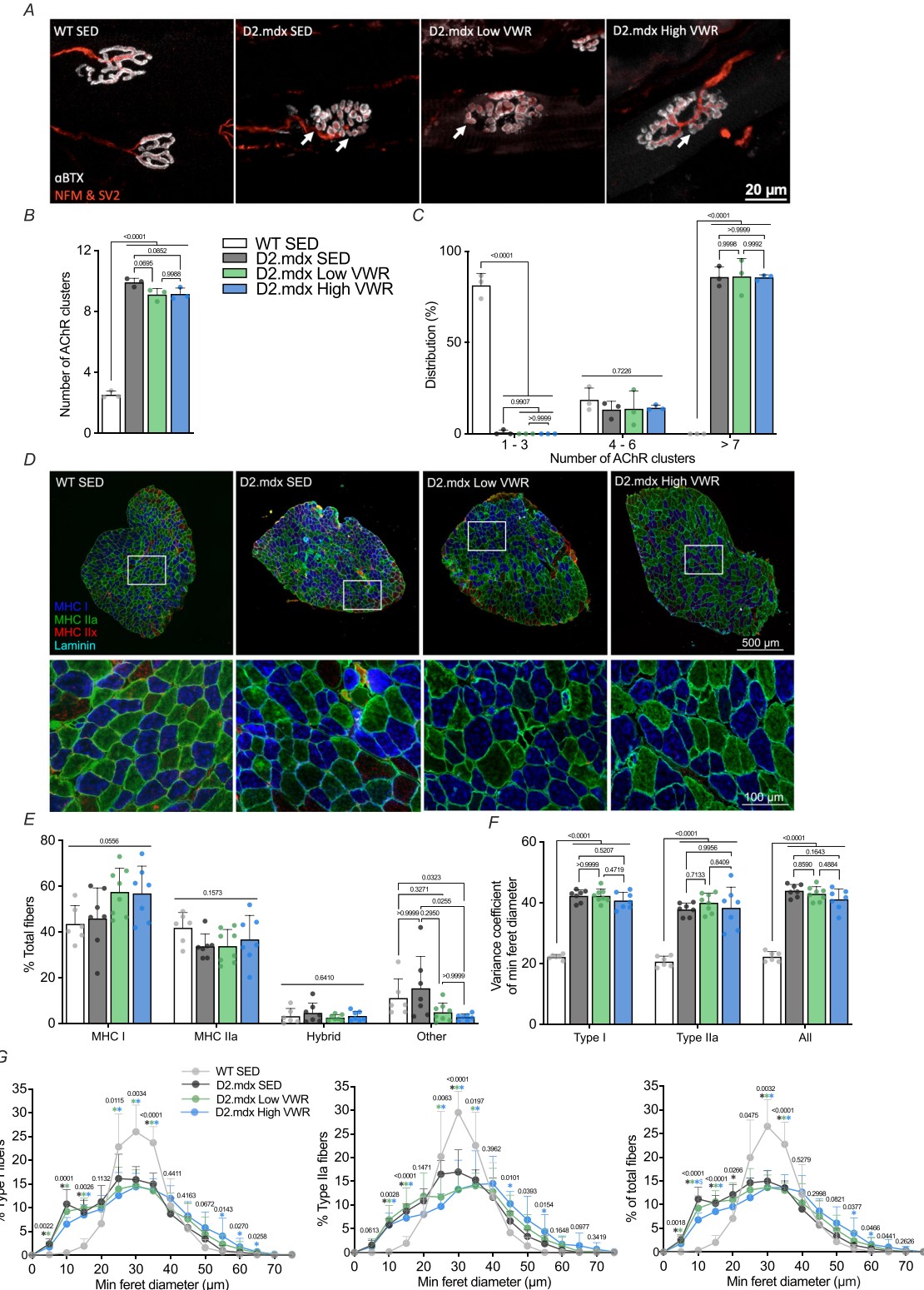

**Figure 4. Voluntary wheel running (VWR) does not compromise neuromuscular junction morphology and shifts muscle fibre type distribution and minimum feret diameter**

*A*, representative confocal immunofluorescence (IF) images of neuromuscular junctions (NMJs) in the epitrochleoanconeus muscle of WT SED, D2.mdx SED, D2.mdx Low VWR and D2.mdx High VWR animals. Pre-synaptic regions are stained red with neurofilament M (NFM) and synaptic vesicle 2 (SV2), whereas the postsynapse is stained with **α**-bungarotoxin (**α**BTX, white). Arrows indicate acetylcholine receptor (AChR) clusters. Scale bar = 20

μm. Quantification of the (*B*) number and (*C*) distribution of AChR clusters per NMJ. *n* = 2–3, with 20–40 NMJs analysed per sample. *D*, representative IF images of soleus (SOL) muscles, including myosin heavy chain (MHC) types I (blue), IIa (green) and IIx (red). Type IIb fibres remain unstained. Scale bar = 500 μm for whole-muscle cross-sections (top panel) and 100 μm for insets (bottom panel). *E*, graphical summary of fibre type distribution of type I, type IIa, hybrid and other fibre types (e.g. type IIx, type IIb, embryonic MHC (eMHC) fibres) in WT SED, D2.mdx SED, D2.mdx Low VWR and D2.mdx High VWR animals. *F*, variance coefficient and (*G*) frequency distribution of minimum (min) feret diameter in type I fibres, type IIa fibres and total fibres. *n* = 6–8. Data are presented as mean ± SD. *$p < 0.05$ *vs*. WT SED and $p < 0.05$ *vs*. D2.mdx SED. For (*B*), (*C*), (*E*–*G*) a one-way ANOVA or the Kruskal–Wallis test was used, followed by a Tukey's or Dunn's multiple comparisons test where appropriate.

significantly increased mitochondrial fission protein 1 (FIS1) expression as a marker of mitochondrial fission in D2.mdx animals *versus* WT controls, with no effect of exercise in dystrophic mice (Fig. 5*D* and *K*). The expression of optic atrophy 1 (OPA1), a driver of mitochondrial fusion, was significantly higher in D2.mdx High VWR animals compared to both WT SED and D2.mdx SED groups (Fig. 5*D* and *L*). Finally, mitofusin-2 (MFN2) expression remained similar between D2.mdx groups but was significantly higher compared to healthy WT controls (Fig. 5*D* and *M*).

## Discussion

The purpose of the current study was to investigate the effects of volitional exercise on muscle structural, functional and molecular phenotypes in the clinically relevant D2.mdx mouse model of DMD. The data demonstrate that a relatively high volume of VWR selectively augmented skeletal muscle mass, particularly in the QUAD, GAST and SOL muscles. This was accompanied by improved muscle function in dystrophic high-volume runners, where locomotor and exploratory behaviour was partially restored, and *ex vivo* muscle force production was improved at varying stimulation frequencies and during taxing ECCs. Furthermore, D2.mdx mice that exercised presented with less fibrotic infiltrate compared to SED, genotype-matched controls, and importantly, VWR did not exacerbate skeletal, cardiac or respiratory muscle quality and integrity. We also observed exercise-induced molecular adaptations in D2.mdx animals that are typically documented in healthy, non-dystrophic skeletal muscle, such as a shift towards a more oxidative fibre type composition, including improved mitochondrial biology and function (Furrer et al., 2023). Altogether, these data indicate that despite the presence of advanced dystrophy in D2.mdx mice, daily volitional physical activity improves muscle quality, and more specifically, high-volume exercise elicits superior benefits compared to a lower alternative. Our results are consistent with the balance of reports examining exercise in the more conventional, but less severe, C57.mdx model of DMD (Delacroix et al., 2018; Hamm et al., 2021; Hyzewicz et al., 2015; Kogelman et al., 2018; Ng et al., 2018; Sigoli et al., 2022; Zhou et al., 2024) and

further support clinical studies (Alemdaroğlu et al., 2015; Anandan et al., 2024; Bulut et al., 2022; Jansen et al., 2013; Lott et al., 2021; Sherief et al., 2021) demonstrating safety and efficacy of informed exercise prescription in DMD patients.

Glucocorticoids are prescribed as a standard of care for DMD patients to slow disease progression; however no cure is currently available. Recently developed corticosteroid compounds, such as vamorolone, show efficacy as treatment options with less adverse side effects than prednisone and deflazacort (Hoffman et al., 2019; Roberts et al., 2023). Molecular therapies that have achieved health and safety regulatory approval in some jurisdictions, such as exon-skipping, stop codon readthrough and gene replacement strategies, work to partially restore dystrophin expression and shift the phenotype to a milder form of dystrophy in patients (Roberts et al., 2023). Collectively, these pharmacological approaches for DMD may demonstrate additive or synergistic benefits when combined with lifestyle-based interventions, such as exercise (Hamm et al., 2021). In the current study, male D2.mdx mice completed a considerable amount of volitional running wheel exercise, averaging 1.18 km/day across all runners, and more specifically 1.84 km/day and 0.46 km/day in the D2.mdx High and Low VWR groups, respectively. This is consistent with volitional running distances in female D2.mdx mice (Monceau et al., 2022) but significantly less than what is typically observed in C57.mdx VWR animals (Hyzewicz et al., 2015), with the difference reasonably attributed to the more severe phenotype in D2.mdx mice (Swiderski & Lynch, 2021). Comparatively, we observed that on average, both the D2.mdx Low and High VWR cohorts completed greater daily running distances compared to those reported in a recent D2.mdx treadmill exercise study by Zelikovich et al. (Zelikovich et al., 2019), which translates into a higher habitual dose of exercise stimulus for our VWR protocol. The relative strengths and weaknesses of VWR *versus* treadmill exercise paradigms in preclinical studies have been summarized elsewhere (Schmitt et al., 2020). Although we cannot appropriately set exercise prescriptions with VWR, it may be the preferred exercise modality to promote beneficial adaptations without exacerbating the dystrophic pathology compared to forced treadmill exercise (Spaulding & Selsby, 2018).

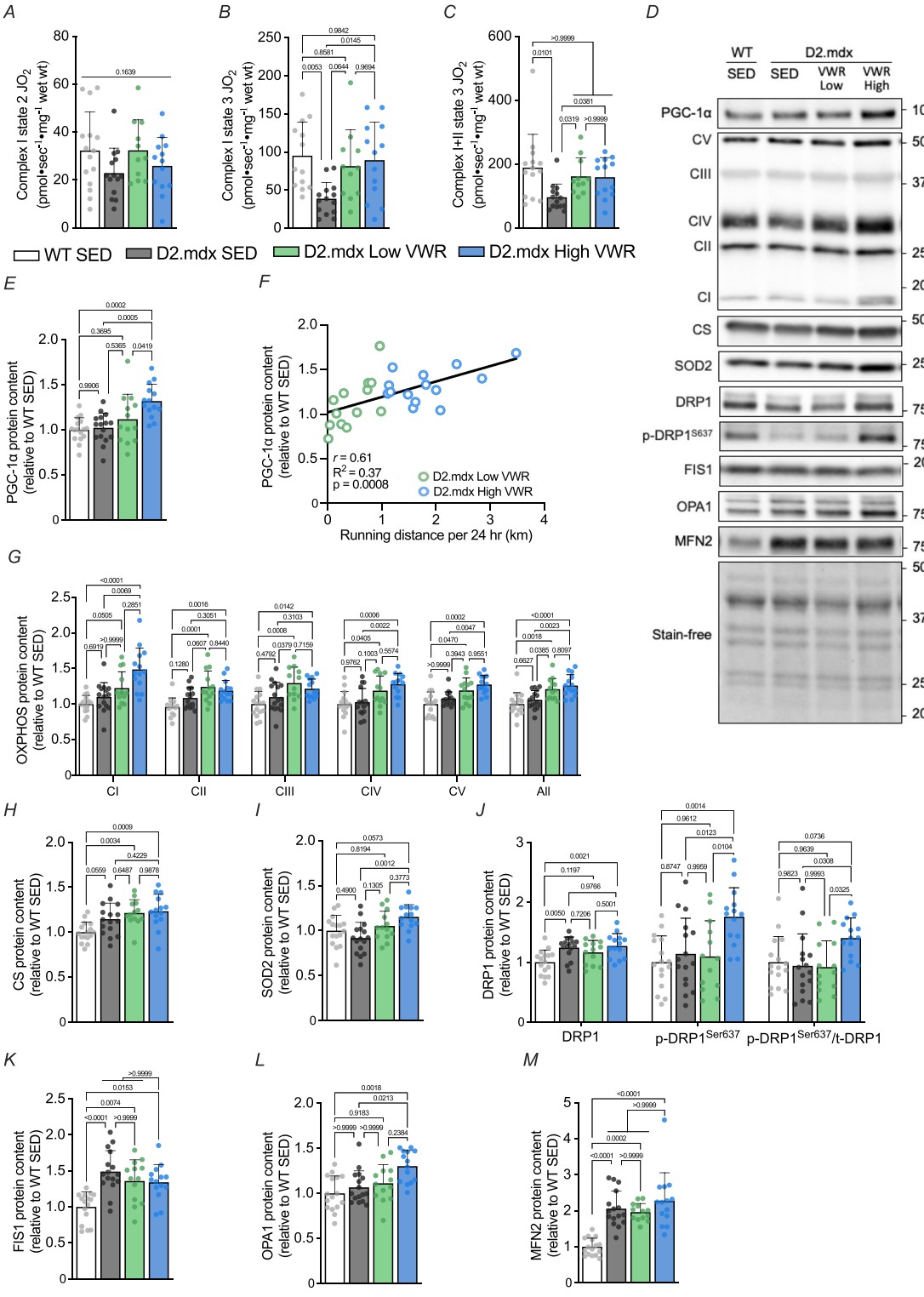

**Figure 5. A relatively high volume of voluntary wheel running (VWR) improves skeletal muscle mitochondrial respiration and augments mitochondrial content and fusion protein expression in D2.mdx mice**

*A*, mitochondrial complex I state 2 respiration (picomoles of oxygen per second per mg wet weight of muscle; pmol·sec-1·mg-1 wet weight) with the addition of pyruvate and malate (PM) as substrates in the quadriceps (QUAD) muscles from animals across the four experimental groups. *B*, complex I state 3 maximal

respiration (pmol·sec-1·mg-1 wet weight) in the presence of PM, plus ADP and glutamate (PMDG), and (*C*) complex I+II state 3 maximal respiration (pmol·sec-1·mg-1 wet weight) induced in the presence of PMDG and succinate. *D*, representative Western blots of peroxisome proliferator-activated receptor-$\gamma$ coactivator-1$\alpha$ (PGC-1$\alpha$), oxidative phosphorylation (OXPHOS) complexes I-V (CI-CV), citrate synthase, (CS), superoxide dismutase 2 (SOD2), dynamin-related protein 1 (DRP1), serine 637 phosphorylated DRP1 (p-DRP1[Ser637]), mitochondrial fission protein 1 (FIS1), optic atrophy 1 (OPA1) and mitofusin-2 (MFN2) in the gastrocnemius (GAST) muscles of WT and D2.mdx animals. Approximate molecular weights (kDa) of the protein ladder are expressed on the right side of blots. Representative stain-free images are shown to indicate sample loading. *E*, graphical summary of PGC-1$\alpha$ protein content and (*F*) correlation with daily running distance in D2.mdx animals. Quantification of (*G*) OXPHOS, (*H*) CS, (*I*) SOD2, (*J*) DRP1, p-DRP1[Ser637] and DRP1 phosphorylation status (i.e. the phosphorylated form of the protein relative to its total content), (*K*) FIS1, (*L*) OPA1 and (*M*) MFN2 protein content following the intervention. Western blot data are expressed relative to the WT SED group. $n$ = 11–16. Data are presented as mean $\pm$ SD. For (*A*), a one-way ANOVA was used. For (*B*), (*C*), (*E*), (*G–M*) a one-way ANOVA or the Kruskal–Wallis test was used, followed by a Tukey's or Dunn's multiple comparisons test where appropriate. For (*F*), a simple linear regression was used.

Our data demonstrate that a relatively high volume of VWR was sufficient to promote physiological adaptations in D2.mdx mice, as evidenced by the partial restoration of skeletal muscle mass and relative force production. Volitional exercise also did not alter the number of CNFs and did not worsen skeletal muscle membrane integrity. Fibrosis was also mitigated in both D2.mdx Low and High VWR groups compared to their SED counterparts. Similarly, despite the severe phenotype in dystrophic respiratory muscles, CNFs remained unchanged, and fibrosis was reduced in the DIA muscle following high-volume VWR compared to D2.mdx SED animals. Additionally, we observed exercise-induced cardiac remodelling with an increased heart mass in D2.mdx High VWR animals compared to untrained WT mice, without changes in the fibrotic profile. These structural adaptations are in support of the improved cardiac fitness typically observed with long-term, aerobic-type exercise (Allen et al., 2001). Together, these data showing multisystemic improvements in the pathophysiology of dystrophic muscle may hold translational importance, especially because cardiac and respiratory muscles are most affected in DMD patients, and exercise-induced cardiorespiratory activity improves patient health (Jansen et al., 2013).

DMD causes structural alterations within the post-synaptic compartment, resulting in compromised NMJ structure and fibre size heterogeneity (Duan et al., 2021; Ng & Ljubicic, 2020). We selected the ETA muscle to examine NMJ morphology because of its thin, sheet-like composition, which is ideal for wholemount staining (Villarroel-Campos et al., 2022). We then found that, although genotype differences in postsynaptic AChR clustering were apparent as expected (Ng & Ljubicic, 2020), VWR did not alter AChRs in the dystrophic ETA. Although VWR has previously been shown to mitigate age-related declines in NMJ morphology (Cheng et al., 2013; Valdez et al., 2010), the lack of an exercise effect in our study may be due to the much more severe myopathy in the D2.mdx model. In addition, similar to previous VWR literature (Bittel et al., 2024; Manzanares et al., 2019; Vanlieshout et al., 2024), we observed a modest shift towards a slower, more oxidative myofiber composition, which is resistant to fatigue. Thus low or high volumes of volitional exercise did not worsen the neuromuscular phenotype in D2.mdx mice, but rather led to improvements in specific force and fatigue resistance, likely attributed to adaptations within the myofiber.

Mitochondrial stress is a common feature in dystrophic animals and DMD patients, which presents as impaired organelle turnover and dynamics, as well as compromised bioenergetic capacity (Mikhail et al., 2023). We found that VWR in D2.mdx animals restored skeletal muscle mitochondrial respiration similar to healthy WT levels compared to SED dystrophic controls. In line with this, mitochondrial content was augmented following high-volume VWR, as evidenced by increased expression of representative subunits of OXPHOS complexes, CS and PGC-1$\alpha$, which was also significantly correlated to running volume. These data are consistent with the shift towards a slow, oxidative myofiber composition following VWR in our study, which is more resistant to the dystrophic pathology (Boyer et al., 2019; Ljubicic et al., 2011). Furthermore, VWR-induced mitochondrial biogenesis was paralleled by increased SOD2 protein content as a marker of mitochondrial antioxidant defence in D2.mdx High VWR animals. Along these lines, future studies should investigate the effects of long-term volitional exercise on oxidative stress, which is elevated in D2.mdx mice (Cleverdon et al., 2022; Hughes et al., 2019; Mikhail et al., 2023; Ramos et al., 2020). After a relatively high volume of VWR in dystrophic animals, we also observed augmented markers of mitochondrial fusion, including OPA1 and phosphorylation at the serine 637 inhibitory mark on DRP1. These findings are in agreement with several studies that investigate mitochondrial dynamics following chronic physical activity (Axelrod et al., 2019; Iqbal et al., 2013). Previous work has also revealed the necessity for OXPHOS to meet the metabolic demand required for fusion events. Specifically, mitochondrial fusion mediated by OPA1 has been coupled to increases in OXPHOS (Mishra

et al., 2014; Yao et al., 2019), similar to the changes we observe in this study. Although this relationship was not directly investigated in our work, we postulate that higher volumes of VWR may enhance mitochondrial biology in dystrophic muscle, partially through a shift in the balance of organelle dynamics towards more fusion. Future studies should employ gold-standard measures, such as electron microscopy, to confirm these morphological pro-fusion adaptations within the mitochondrial reticulum following VWR in dystrophic muscle.

We acknowledge a few limitations in this work. First, although the time frame of this study was carefully chosen to capture the progression of skeletal muscle degeneration in D2.mdx animals, it represents earlier stages of cardiac remodelling (Coley et al., 2016; Hayes et al., 2022; Hughes et al., 2020; Kennedy et al., 2018; van Putten et al., 2019). Histopathological changes in cardiac tissue become evident around 14–21 weeks of age in D2.mdx mice, whereas gene expression and bioenergetic capacity are impaired at earlier time points (Hughes et al., 2020; Kennedy et al., 2018). Thus the onset of cardiac remodelling likely takes place towards the end of our experimental protocol when mice are 15–18 weeks of age. Future studies should examine a later time point of the D2.mdx cardiomyopathy to further characterize the effects of VWR in the dystrophic heart. Furthermore, we noted large variability in free VWR behaviour of the D2.mdx animals, which was to be expected to some extent (Brown et al., 2022; O'Neal et al., 2017). However, this was despite controlling for age and sex of the mice, as well as the environmental conditions during the study. Finally, although all D2.mdx animals were randomized into SED and VWR groups, neither baseline measurements of *in vivo* muscle function and body composition were conducted to confirm that both cohorts were similar, nor were longitudinal assessments performed to monitor their condition over the course of the study. Although individual D2.mdx mice exhibit variability in dystrophic phenotype severity and some differences exist in the motivation to exercise in the Low and High VWR groups, we speculate that D2.mdx Low VWR animals were not limited by a more severe phenotype, as several histological markers remained similar to or improved compared to D2.mdx SED animals.

In conclusion our data demonstrate that endurance-type volitional exercise augments specific muscle function, attenuates skeletal and respiratory muscle degeneration and enhances mitochondrial biology in the relatively severe D2.mdx model of DMD. We note that despite lower VWR distance in D2.mdx mice compared to typical C57.mdx animals, we observed dose–response improvements with low and high volumes of exercise. Importantly, there was no evidence of VWR exacerbating the dystrophic pathology. Future research should explore the acute signalling events that contribute to these exercise adaptations, particularly given that dystrophic muscle may be less responsive to contraction-induced mechanical stimuli (Boccanegra et al., 2023). Thus, using a more clinically meaningful model of DMD, these findings provide further preclinical support for exercise as a therapeutic modality. However, it remains uncertain whether these improvements will translate to boys with DMD, where the clinical and functional presentation of the disease is much more severe. Continued examination of exercise biology in DMD, as well as in combination with novel pharmacological therapeutics, is critical for identifying effective treatments that may enhance the quality of life in patients.

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

## Additional information

### Data availability statement

All data within the current study are available from the corresponding author upon reasonable request.

### Competing interests

No competing interests declared.

### Author contributions

S.R.M. and V.L. conceived and designed the experiments. S.R.M., S.Y.N. and A.I.M. collected the samples. S.R.M., S.Y.N., A.I.M., D.W.S., C.E.J. and I.A.R. ran experiments and analysed the data. S.R.M. and V.L. interpreted results of the experiments. S.R.M. prepared the figures. S.R.M. and V.L. drafted the manuscript or revised it critically for important intellectual content. All authors have read and approved the final version of this manuscript and agree to be accountable for all aspects of the work in ensuring that questions related to the accuracy or integrity of any part of the work are appropriately investigated and resolved. All persons designated as authors qualify for authorship, and all those who quality for authorship are listed.

### Funding

This work was supported by the Canadian Institutes of Health Research, the Canada Research Chairs program, the Ontario Ministry of Economic Development, Job Creation and Trade (MEDJCT) and Muscular Dystrophy Canada. S.R.M., S.Y.N. and D.W.S. are postgraduate scholars supported by the Natural Science and Engineering Council of Canada. A.I.M. is an Ontario Graduate scholar. V.L. is the Canada Research Chair (tier 2) in neuromuscular plasticity in health and disease and is a recipient of the MEDJCT Early Researcher Award.

### Keywords

exercise, mitochondria, neuromuscular disorders, skeletal muscle

## Supporting information

Additional supporting information can be found online in the Supporting Information section at the end of the HTML view of the article. Supporting information files available:

**Peer Review History**

