## [Peer Review History · The Journal of Physiology]

Volitional exercise elicits physiological and molecular improvements in the severe D2.mdx mouse model of Duchenne muscular dystrophy

Stephanie R Mattina, Sean Y Ng, Andrew I Mikhail, Derek W Stouth, Cora Esmenia Jornacion, Irena A Rebalka, Thomas J Hawke, and Vladimir Ljubcic

DOI: 10.1113/JP286768

Corresponding author(s): Vladimir Ljubcic (ljubicic@mcmaster.ca)

Review Timeline:

Submission Date:	16-Apr-2024
Editorial Decision:	29-May-2024
Revision Received:	17-Dec-2024
Editorial Decision:	09-Jan-2025
Revision Received:	23-Jan-2025
Accepted:	27-Jan-2025

Senior Editor: Richard Carson

Reviewing Editor: Bruno Grassi

Transaction Report:

Dear Dr Ljubicic,

Re: JP-RP-2024-286768 "Long-term, daily volitional exercise elicits physiological and molecular improvements in the severe D2.mdx mouse model of Duchenne muscular dystrophy" by Stephanie R Mattina, Sean Y Ng, Andrew I Mikhail, Derek W Stouth, Cora Jornacion, Irena Rebalka, Thomas J Hawke, and Vladimir Ljubicic

Thank you for submitting your manuscript to The Journal of Physiology. It has been assessed by a Reviewing Editor and by 2 expert referees and we are pleased to tell you that it is potentially acceptable for publication following satisfactory major revision.

REVISION CHECKLIST:

Please upload two versions of your manuscript text: one with all relevant changes highlighted and one clean version with no changes tracked. The manuscript file should include all tables and figure legends, but each figure/graph should be uploaded as separate, high-resolution files. The journal is now integrated with Wiley's Image Checking service. For further details, see: <https://www.wiley.com/en-us/network/publishing/research-publishing/trending-stories/upholding-image-integrity-wileys->

image-screening-service

We look forward to receiving your revised submission.

Yours sincerely,

Richard Carson
Senior Editor
The Journal of Physiology

REQUIRED ITEMS FOR REVISION

- Author photo and profile. First or joint first authors are asked to provide a short biography (no more than 100 words for one author or 150 words in total for joint first authors) and a portrait photograph. These should be uploaded and clearly labelled together in a Word document with the revised version of the manuscript. See Information for Authors for further details.

- You must start the Methods section with a paragraph headed Ethical Approval. A detailed explanation of journal policy and regulations on animal experimentation is given in Principles and standards for reporting animal experiments in The Journal of Physiology and Experimental Physiology by David Grundy *J Physiol*, 593: 2547-2549. doi:10.1113/JP270818). A checklist outlining these requirements and detailing the information that must be provided in the paper can be found at: <https://physoc.onlinelibrary.wiley.com/hub/animal-experiments>. Authors should confirm in their Methods section that their experiments were carried out according to the guidelines laid down by their institution's animal welfare committee, and conform to the principles and regulations as described in the Editorial by Grundy (2015), including an ethics approval reference number. The Methods section must contain a statement about access to food, water and housing, details of the anaesthetic regime: anaesthetic used, dose and route of administration, and method of killing the experimental animals.

- You must upload original, uncropped western blot/gel images (including controls) if they are not included in the manuscript. This is to confirm that no inappropriate, unethical or misleading image manipulation has occurred. These should be uploaded as 'Supporting information for review process only'. Please label/highlight the original gels so that we can clearly see which sections/lanes have been used in the manuscript figures. For more information, see: <https://physoc.onlinelibrary.wiley.com/hub/journal-policies#imagmanip>.

- Papers must comply with the Statistics Policy: https://jp.msubmit.net/cgi-bin/main.plex?form_type=display_requirements#statistics.

In summary:

- If $n \leq 30$, all data points must be plotted in the figure in a way that reveals their range and distribution. A bar graph with data points overlaid, a box and whisker plot or a violin plot (preferably with data points included) are acceptable formats.

- If $n > 30$, then the entire raw dataset must be made available either as supporting information, or hosted on a not-for-profit repository, e.g. FigShare, with access details provided in the manuscript.

- 'n' clearly defined (e.g. x cells from y slices in z animals) in the Methods. Authors should be mindful of pseudoreplication.

- All relevant 'n' values must be clearly stated in the main text, figures and tables.

- The most appropriate summary statistic (e.g. mean or median and standard deviation) must be used. Standard Error of the Mean (SEM) alone is not permitted.

- Exact p values must be stated. Authors must not use 'greater than' or 'less than'. Exact p values must be stated to three significant figures even when 'no statistical significance' is claimed.

- Please include an Abstract Figure file, as well as the Figure Legend text within the main article file. The Abstract Figure is a piece of artwork designed to give readers an immediate understanding of the research and should summarise the main conclusions. If possible, the image should be easily 'readable' from left to right or top to bottom. It should show the physiological relevance of the manuscript so readers can assess the importance and content of its findings. Abstract Figures should not merely recapitulate other figures in the manuscript. Please try to keep the diagram as simple as possible and without superfluous information that may distract from the main conclusion(s). Abstract Figures must be provided by authors no later than the revised manuscript stage and should be uploaded as a separate file during online submission labelled as File Type 'Abstract Figure'. Please also ensure that you include the figure legend in the main article file. All Abstract Figures should be created using BioRender. Authors should use The Journal's premium BioRender account to export high-resolution images. Details on how to use and access the premium account are included as part of this email.

EDITOR COMMENTS

Reviewing Editor: Comments for Authors to ensure the paper complies with the Statistics Policy (Required):
SD should be used instead of SE.

Comments to the Author (Required):

The present is a comprehensive study, carried out by a multidisciplinary approach, aimed at investigating a relevant issue in the muscular dystrophy field, that is whether in Duchenne muscular dystrophy exercise can be performed without exacerbating muscle injury and damage. The study provides further evidence of benefits of aerobic exercise in dystrophic mice. The Reviewers raise several comments, which the authors should take in due consideration. More specifically, Reviewer 1 points to the limited baseline measurements, to data heterogeneity, to the lack of a WT training group, to the issue of more clear discussion of differences between mouse and human disease. Reviewer 2 raises several methodological and data interpretation considerations and weaknesses of the study, related for example to the issues of mitochondrial respiration and fission, histology. etc. Also Reviewer 2 suggests to tune down the clinical implications in the extrapolation of the findings to human disease.

REFeree COMMENTS

Referee #1:

This is a comprehensive manuscript that provides further evidence of the benefits of aerobic exercise in dystrophic mice. The study extends the body of work showing benefits of regular exercise, including on mitochondria respiration/content, force measures, and reduces fibrosis in a more severe mdx model, D2.mdx. Given the questions on the benefits/detriments of exercise in muscular dystrophies, the study has the potential to have a high impact. However, I have several comments and questions.

1) Since the mdx exercise training groups were divided into two groups after the training based on how far they ran, it can not be ruled out that the mice in the high running volume had improved phenotype (e.g., mitochondria function) prior to the study, and that may have been the reason they performed better after training. Limited baseline measures before training are provided. With this study design, it can not be definitely determined that the high volume training elicited superior benefits compared to a lower volume training. It may have been that those who had more favorable phenotype before training, did more running during training.

2) Could any possible explanations be provided for the considerable heterogeneity in distance run during training among mdx mice?

3) Another limitation of the study not acknowledged is that there is no WT exercise group. It is not clear whether the

adaptations were similar to unaffected mice.

4) 8-10 weeks of exercise training does not seem to be "long term". Particularly in light of a recent study evaluating exercise training in dystrophic mice over 52 weeks (Hamm et al. *Front Physiol.*, 26:14:1166206, 2023). Long term should be changed in title and elsewhere in manuscript.

5) While the D2.mdx model is a more severe phenotype, there are still significant differences to the human disease progression (significantly less fatty tissue accumulation) that should be acknowledged. For example, multiple studies (Hammers et al. *Sci Rep.* 10: 14070, 2020; Coley et al. *Hum Mol Genet.* 25(1):130-45, 2016) have reported substantial calcification in the D2.mdx model.

6) While the findings are encouraging, translating these findings to humans is premature. There are several notable differences with the muscle in D2. mdx mice and human muscle, and the humans with DMD appear to have a much more severe clinical/functional phenotype - lose ambulation at a young age, etc. Therefore, statements such as "Informed exercise prescription is an intervention that is suitable for all patients..... will aid in enhancing quality of life in patients." need to be tempered when discussing translation of these findings to humans.

7) Any possible explanation for the greater relative composition of type I fiber composition in mdx mice that trained? Could this be due to an accelerated decline in type II fiber due to muscle damage in mdx with training? Could a reference be provided showing fiber type changes being expected within 8-10 weeks of wheel running in wild-type mice?

8) line ~ 705 It is not clear how this study is relevant to resistance training in DMD. No strength training was done in these mice. This part of the discussion should be considered to be omitted.

Referee #2:

The manuscript of Mattina et al., deals with an interesting topic, i.e. the assessment of voluntary wheel running on the pathology progression of D2 mdx mice, having a more severe phenotype with respect to the classical C57/mdx. The authors present a large amount of data from a multidisciplinary approach, based on the bimodal distribution of D2 mdx mice to perform a voluntary high or low wheel exercise and describing a sort of "dose-related" beneficial effects on some functional, histological and metabolic endpoints.

The manuscript is of interest for the field, considering that the threshold between benefit and damage by exercise in DMD-pathology is rather narrow, compelling the interest of research on exercise to understand mechanisms underlying potential benefits by low intensity physical activity in DMD patients.

While the manuscript is of interest, I envisage main point of weakness, both in protocol and data presentation and interpretation, that would require attention.

Below a list of my concerns:

1) There are many methodological weaknesses throughout the manuscript, as follow:

a) the gender of the mice should be disclosed, especially considering the different exercise susceptibility between genders, as well as the number of animals per experimental group. A scheme of the study design may help. Please comment and describe if protocols are adherent to TREAT-NMD SOP for preclinical studies in DMD.

b) it is stated "D2.mdx VWR animals were individually housed with access to a home-cage running wheel (Columbus Instruments, Columbus, OH, USA) for 8-10 weeks. The number of revolutions were recorded every 10 min for the duration of the intervention". How were revolutions measured? Was there any visual human supervision to be sure that the mouse really performs the exercise (sometime mice play making the wheel turn from outside). I understand that the wheel is in the cage for the entire duration of the protocol (8-10 weeks), so I guess that revolutions were measured every 10 minutes over 24 hours for the entire duration. Is that correct? Please detail.

c) The in vivo parameters were only measured at the end of the VWR protocol. While I appreciate the possibility to compare

the final outcome of training on those parameters, however, we missed the main information on how the chronic voluntary running may influence any possible time-related change.

- d) For ex vivo muscle force the group of wt animals is missing. The reasons for not measuring or for not showing this key functional parameter should be explained, otherwise data have to be provided to better appreciate the effect of high VWR and the recovery score.
- e) The protocol of treadmill running is confusing as it states. I understand that is made to assess if the ETA muscle responds to exercise such as TA one, in view of the following experiment on NM junction; however this part is not informative in the whole context. Maybe can be moved in the supplementary material. Otherwise I would suggest a comparison with the D2-mdx, that may add important piece of information in relation to the ability to properly respond to a more intense exercise (see below).
- f) The statistical analysis is not clear in the term of hypothesis to be tested and when one or two way ANOVA was used. This should be made clearer in the methods and throughout the manuscript.
- g) The assessment of the amount of distance run is also confusing. The first weeks are characterized by a progressive increase in the distance ran, in all animals. The calculation of a mean daily value from all mice at all time points is a miscalculation. First, it should be verified that this mean value follows a normal distribution, which I doubt considering the presence of two population of runners. I think that a more correct way would be to show the median daily value around the plateau and then show data for their gaussian distributions at the various time points, which would have allowed to have a more precise definition of the size of the two populations. In the present form I think that an important bias might have been introduced in the separation of the two groups. The assessment of running attitude of wt animals would have been highly informative considering that the D2 wt animals also have a phenotype.

2) The manuscript is not really focused on the threshold of mechanical activation required to have adaptative changes. The described changes in mitochondrial "function" is of interest. However, some key players are missing. For instance, pAMPK has only be measured to assess that ETA muscle of wt animals correctly respond to an acute exercise bout. In contrast, the effect of low or high VWR on the main mechano-metabolic signaling, i.e. pAMPK has not be assessed, although AMPK plays a key role in PGC-1 alpha activation and mitochondrial biogenesis. This is an important point, since most of the damaging effect of chronic forced exercise has been related to the inability of dystrophic muscle to adequately "sense" and adapt to new mechanical requirement. While I do agree that voluntary exercise may have a different outcome vs forced one and might exert benefit vs damage, the two approaches are important to actually understand the threshold of activity and need to be seen as a whole. In this regard, part of key literature is missing or not considered. For instance, basal pAMPK has been found to be higher in both C57mdx and D2 mdx, suggesting an overactivation of this signaling (already at normal cage activity) likely related to the altered mechano-transduction in dystrophin deficient muscle (Boccanegra et al DMM 2023), with minor changes exerted by treadmill exercise. I suggest to verify the state of AMPK activation in all the present conditions and to comment the results in line of the available evidences that need to be properly quoted.

3) The paragraph on mitochondrial respiration and the other on mitochondrial fusion needs to be revised according to the data. In many cases, no phenotypic differences are present between wt and sedentary D2mdx (see PGC 1a, OXPHOS proteins, OPA1 etc) then the change by VWR in terms of "beneficial" is not clear. By contrast, in other cases the phenotype is clear, but no effect of VWR is observed. Attention is needed to not over-interpret the results not to discuss trends in these two paragraphs and generally throughout the manuscript.

4) For histology: a more direct marker of myonecrosis and inflammation should be provided. Also it is not clear if morphometric analysis has been done on the entire section (as correct) or on selected areas.

5) The discussion should be revised based on the fact that a VWR of this type does not in fact help to set any specific exercise protocol. The amount of distance ran over the 24h is not a real information on the amount of work, and important info are missing, such as continuity, fatigue, speed etc. Then I would suggest to tune down the clinical implications, as this may be misleading.

Other points

Keywords need to be changed to be more adherent to the aim of the work

In the key points "Volitional exercise normalized and upregulated dystrophic skeletal muscle mitochondrial respiration and content, respectively. Furthermore, a higher relative dose of VWR increased organelle fusion protein expression in D2.mdx animals": rewrite to avoid misinterpretation (higher vs what?)

Introduction lines 99-101: "Recent literature that examined the effects of endurance-type, treadmill run training and a resistance-type synergist ablation hypertrophic model in D2.mdx mice demonstrate improved force production and reduced markers of the dystrophic pathology". The work of Hassani et al has many limitation and cannot be really compared to a physiological exercise protocol. I would suggest to take this reference out as not relevant.

introduction lines 102 up to 107: "In contrast to these forced activity modalities, rodent voluntary wheel running (VWR) mimics the ability of humans to self-regulate exercise" "Further investigation of volitional exercise in DMD could contribute to the development of exercise prescriptions for patients and may uncover therapeutic avenues that modulate the dystrophic pathology" I think this statement is incorrect, the ability to self-regulate exercise can depend on the state of neuromuscular function and to other less controlled variables. As in the present study, the variability can be very high, in humans even worse. I do not think that "volitional exercise" can be translated to real exercise prescription program of physical therapy and is also a rather dangerous message, considering the high expectation of parents, that may push their DMD child to more and more volitional exercise, in spite of fatigue and in the absence of any clear indications. Then I suggest to tune this message down.

Lines 415-416 "Previous research examining the effects of exercise on skeletal muscle integrity and

histology in DMD are inconclusive" I think this sentence is incorrect and not necessary. Surely, there is a different effect of exercise on pathology progression based on the protocol used. But for sure is not the protocol presented herein to allow "conclusive" results.

Body mass for D2 wt ad mdx appear to be rather higher vs values observed in other studies. Please comment.

END OF COMMENTS

Authors' Response to Reviewer Comments:

EDITOR COMMENTS

Reviewing Editor: Comments for Authors to ensure the paper complies with the Statistics Policy (Required):

SD should be used instead of SE.

Comments to the Author (Required):

The present is a comprehensive study, carried out by a multidisciplinary approach, aimed at investigating a relevant issue in the muscular dystrophy field, that is whether in Duchenne muscular dystrophy exercise can be performed without exacerbating muscle injury and damage. The study provides further evidence of benefits of aerobic exercise in dystrophic mice. The Reviewers raise several comments, which the authors should take in due consideration. More specifically, Reviewer 1 points to the limited baseline measurements, to data heterogeneity, to the lack of a WT training group, to the issue of more clear discussion of differences between mouse and human disease. Reviewer 2 raises several methodological and data interpretation considerations and weaknesses of the study, related for example to the issues of mitochondrial respiration and fission, histology. etc. Also Reviewer 2 suggests to tune down the clinical implications in the extrapolation of the findings to human disease.

We kindly thank the Editor for their time and for sharing the Reviewers' insightful feedback. We are grateful for the opportunity to revise our work and address the detailed and fair points that have been raised. Since our previous submission, we have adjusted the manuscript figures to adhere to the Journal of Physiology Statistics Policy. We have carefully considered and responded to the Reviewers' appraisal of our paper and have worked to address these concerns to strengthen our manuscript.

REFEREE COMMENTS

Referee #1:

This is a comprehensive manuscript that provides further evidence of the benefits of aerobic exercise in dystrophic mice. The study extends the body of work showing benefits of regular exercise, including on mitochondria respiration/content, force measures, and reduces fibrosis in a more severe mdx model, D2.mdx. Given the questions on the benefits/detriments of exercise in muscular dystrophies, the study has the potential to have a high impact. However, I have several comments and questions.

1) Since the mdx exercise training groups were divided into two groups after the training based on how far they ran, it can not be ruled out that the mice in the high running volume had improved phenotype (e.g., mitochondria function) prior to the study, and that may have been the

reason they performed better after training. Limited baseline measures before training are provided. With this study design, it can not be definitely determined that the high volume training elicited superior benefits compared to a lower volume training. It may have been that those who had more favorable phenotype before training, did more running during training.

We thank the Reviewer for their input and agree that this is a limitation that should be made clearer. Due to the invasive nature of the experiments that assess mitochondrial function and content, it was not feasible to measure these prior to the study. However, for the few baseline measures that were available, but unfortunately not included at the start of the study (i.e., body weight, body composition, grip strength, open field activity), there were no observed differences between our D2.mdx SED and VWR groups post-intervention. While these available baseline measures likely would not have influenced our findings, we certainly acknowledge this as a limitation while interpreting the results. We have now emphasized this important point in our discussion of the limitations of the present study (pg. 20, lines 594-597).

2) Could any possible explanations be provided for the considerable heterogeneity in distance run during training among mdx mice?

This is a great question that we have considered, but do not yet fully understand. In our study, the D2.mdx mice were age- and sex-matched, and we heavily controlled for external factors, such as housing and temperature. We speculate that the heterogeneity in running distance may reflect differences in motivation, energy levels, or subtle epigenetic variation. Notably, we observed that in our D2.mdx model, the mice that ran relatively lower volumes were not “worse off” in any of our measured outcomes compared to D2.mdx SED animals. Thus, we postulate that these mice were not at a physical disadvantage prior to the study, however as noted above, we could not assess these baseline measures. Nonetheless, this heterogeneity in running distance is consistent with VWR literature showing considerable variability (O’Neal et al. Curr Biol. 27(3), 423-430, 2017; Brown et al. J Appl Physiol. 132(3):824-834, 2022). We have addressed these points in our statements on the limitations of our study, on pg. 20, lines 591-594 and 598-602.

3) Another limitation of the study not acknowledged is that there is no WT exercise group. It is not clear whether the adaptations were similar to unaffected mice.

We appreciate this point from the Reviewer. We acknowledge that the addition of a WT exercise group would indeed allow for additional comparative data, however, the main focus of our study was to evaluate the effects of exercise in the dystrophic condition. A large body of literature already exists demonstrating the effects of exercise in unaffected mice, and we do not believe that including this group would address our research question. Nevertheless, we make some descriptive comparisons to this WT VWR literature, for example pg. 19, lines 550-552. We have also revised our Methods section to acknowledge the absence of a WT exercise group in our study and referenced a recent, comprehensive review (Smith et al. Nat Rev Mol Cell Biol. 24(9):607-632, 2023) that explores exercise adaptations in healthy animals (pg. 5, lines 140-141).

4) 8-10 weeks of exercise training does not seem to be "long term". Particularly in light of a recent study evaluating exercise training in dystrophic mice over 52 weeks (Hamm et al. *Front Physiol.*, 26:14:1166206, 2023). Long term should be changed in title and elsewhere in manuscript.

We value the Reviewers feedback and have adjusted this in the title and throughout the document.

5) While the D2.mdx model is a more severe phenotype, there are still significant differences to the human disease progression (significantly less fatty tissue accumulation) that should be acknowledged. For example, multiple studies (Hammers et al. *Sci Rep.* 10: 14070, 2020; Coley et al. *Hum Mol Genet.* 25(1):130-45, 2016) have reported substantial calcification in the D2.mdx model.

We thank the Reviewer for pointing out these important differences. These distinctions have now been incorporated into the Methods section of the manuscript (pg. 5, lines 128-132).

6) While the findings are encouraging, translating these findings to humans is premature. There are several notable differences with the muscle in D2. mdx mice and human muscle, and the humans with DMD appear to have a much more severe clinical/functional phenotype - lose ambulation at a young age, etc. Therefore, statements such as "Informed exercise prescription is an intervention that is suitable for all patients..... will aid in enhancing quality of life in patients." need to be tempered when discussing translation of these findings to humans.

We appreciate the Reviewers comment and believe that these findings should be interpreted with more caution. We have modified these statements accordingly (pg. 21, lines 612-616).

7) Any possible explanation for the greater relative composition of type I fiber composition in mdx mice that trained? Could this be due to an accelerated decline in type II fiber due to muscle damage in mdx with training? Could a reference be provided showing fiber type changes being expected within 8-10 weeks of wheel running in wild-type mice?

We believe that the greater relative type I fiber composition reflects an exercise-induced adaptation with our wheel running intervention, where the type II fibers adopt slow, oxidative characteristics. In the literature, changes in fiber type composition have been observed with as little as 4 weeks of wheel running (Vogel et al. *PLoS One.* 10(6):e0130769, 2015), or 6 weeks of wheel running (Bittel et al. *iScience.* 27(1):108632, 2024), with additional studies observing similar changes after 8-10 weeks in WT mice (Pellegrino et al. *Eur J Appl Physiol.* 93(5-6):655-664, 2005; Vanlieshout et al. *Med Sci Sports Exerc.* 56(3):486-498, 2024). It is also possible that the reduced proportion of "Other" fiber types in D2.mdx High VWR vs D2.mdx SED animals may reflect a decrease in type IIb, type IIx, or embryonic MHC+ fibers, suggesting less muscle injury with VWR.

We have provided data below demonstrating that the total number of fibers in the SOL muscles are unaffected across our experimental groups. Thus, we have no evidence that might suggest an accelerated decline in type II fibers following VWR. We have adjusted the manuscript to include more appropriate references showing the expected fiber type shift following VWR (pg. 19, lines 550-552).

8) line ~ 705 It is not clear how this study is relevant to resistance training in DMD. No strength training was done in these mice. This part of the discussion should be considered to be omitted.

The discussion of this reference has been removed.

Referee #2:

The manuscript of Mattina et al., deals with an interesting topic, i.e. the assessment of voluntary wheel running on the pathology progression of D2 mdx mice, having a more severe phenotype with respect to the classical C57/mdx. The authors present a large amount of data from a multidisciplinary approach, based on the bimodal distribution of D2 mdx mice to perform a voluntary high or low wheel exercise and describing a sort of “dose-related” beneficial effects on some functional, histological and metabolic endpoints.

The manuscript is of interest for the field, considering that the threshold between benefit and damage by exercise in DMD-pathology is rather narrow, compelling the interest of research on exercise to understand mechanisms underlying potential benefits by low intensity physical activity in DMD patients.

While the manuscript is of interest, I envisage main point of weakness, both in protocol and data presentation and interpretation, that would require attention.

Below a list of my concerns:

- 1) There are many methodological weaknesses throughout the manuscript, as follow:
 - a) the gender of the mice should be disclosed, especially considering the different exercise

susceptibility between genders, as well as the number of animals per experimental group. A scheme of the study design may help. Please comment and describe if protocols are adherent to TREAT-NMD SOP for preclinical studies in DMD.

We thank the Reviewer for pointing this out. We have revised our Methods section to clarify the sex of the mice and number of animals per experimental group (pg. 5, lines 141-144). Our in vivo functional measures were completed in accordance with the TREAT-NMD SOPs for preclinical studies in DMD, and we have now included this in our Methods section as well (pg. 5, lines 170-171, 175-177).

b) it is stated "D2.mdx VWR animals were individually housed with access to a home-cage running wheel (Columbus Instruments, Columbus, OH, USA) for 8-10 weeks. The number of revolutions were recorded every 10 min for the duration of the intervention". How were revolutions measured? Was there any visual human supervision to be sure that the mouse really performs the exercise (sometime mice play making the wheel turn from outside). I understand that the wheel is in the cage for the entire duration of the protocol (8-10 weeks), so I guess that revolutions were measured every 10 minutes over 24 hours for the entire duration. Is that correct? Please detail.

The revolutions were automatically recorded by the Opto-M3 activity meter on the Multi-Device Interface software on Version 1.5, which was connected to each wheel. The software automatically recorded the number of revolutions every 10 minutes for the entire duration of the intervention. Each of the mice were monitored periodically throughout the intervention to confirm that they were indeed performing the exercise and not spinning the wheel by other means. However, the mice largely perform their exercise at night, and we avoided watching them during this time to prevent any disturbance to their running behaviour. We apologize for the confusion and have further clarified these details in the Methods section on pg. 5, lines 143-149.

c) The in vivo parameters were only measured at the end of the VWR protocol. While I appreciate the possibility to compare the final outcome of training on those parameters, however, we missed the main information on how the chronic voluntary running may influence any possible time-related change.

We thank the reviewer for this feedback. As mentioned in our above response to Reviewer 1 (Comment 1), we have amended our Discussion section to acknowledge this limitation (pg. 20, lines 594-597). Unfortunately, we did not measure in vivo parameters at the start of the study, and in turn, cannot evaluate these time-dependent changes. However, since we also did not observe any differences in these in vivo measures (i.e., body weight, body composition, grip strength, open field activity) between our D2.mdx groups, we are uncertain if including these measures would have influenced our overall findings. Nonetheless, we acknowledge this as an important limitation of our study.

d) For ex vivo muscle force the group of wt animals is missing. The reasons for not measuring or for not showing this key functional parameter should be explained, otherwise data have to be provided to better appreciate the effect of high VWR and the recovery score.

We have excluded the WT SED group from our ex vivo muscle force assessments in order to gain a better appreciation for, and more clearly communicate to the reader, the augmented force production and protection from ECC-induced force drop following High VWR in our D2.mdx animals, specifically. The inclusion of the WT SED group does not change our statistical significance or interpretation of the results, as shown below (A-C). While we hypothesized that the High VWR group would show improvements, we did not expect them to recover to WT levels. Thus, we chose to only display our D2.mdx groups so the WT SED group would not divert attention away from this effect, which we believe to be an important point of emphasis.

e) The protocol of treadmill running is confusing as it states. I understand that is made to assess if the ETA muscle responds to exercise such as TA one, in view of the following experiment on NM junction; however this part is not informative in the whole context. Maybe can be moved in the supplementary material. Otherwise I would suggest a comparison with the D2-mdx, that may add important piece of information in relation to the ability to properly respond to a more intense exercise (see below).

We apologize for the confusion with our treadmill running protocol. We now recognize and agree that this may not be informative in the context of our study. This experiment has been removed from our revised manuscript as per the Reviewers recommendation.

f) The statistical analysis is not clear in the term of hypothesis to be tested and when one or two way ANOVA was used. This should be made clearer in the methods and throughout the

manuscript.

We thank the Reviewer for bringing this forward and apologize for the lack of clarity. We have included additional details of the statistical analyses that were performed in the Methods section (pg. 11-12, lines 318-332). We have also explained our statistical analyses more clearly in our Figure Legends for each graph.

g) The assessment of the amount of distance run is also confusing. The first weeks are characterized by a progressive increase in the distance ran, in all animals. The calculation of a mean daily value from all mice at all time points is a miscalculation. First, it should be verified that this mean value follows a normal distribution, which I doubt considering the presence of two population of runners. I think that a more correct way would be to show the median daily value around the plateau and then show data for their gaussian distributions at the various time points, which would have allowed to have a more precise definition of the size of the two populations. In the present form I think that an important bias might have been introduced in the separation of the two groups. The assessment of running attitude of wt animals would have been highly informative considering that the D2 wt animals also have a phenotype.

We thank the Reviewer for this important consideration. We aimed to report our running distances throughout the intervention in a manner that is consistent with previous literature (Allen et al. J Appl Physiol. 90(5):1900-1908, 2001; Landisch et al. Muscle and Nerve. 38(4):1290-1303, 2008; Rueggsegger et al. J Physiol. 595(1):363-384, 2017; Manta et al. J Physiol. 597(5):1361-1381, 2019) in order to allow for more straightforward comparisons. Several previous reports have demonstrated a large variability in volitional running behaviour in mice (O'Neal et al. Curr Biol. 27(3), 423-430, 2017; Brown et al. J Appl Physiol. 132(3):824-834, 2022). As such, we were expecting similar observations in our D2.mdx animals, and therefore decided to separate the groups to determine if total running volume throughout the 8-10 week intervention would influence the adaptive response. We aimed to address this by comparing relatively low and high volume runners in an impartial and unbiased way, by separating our groups using the median running distance.

The Reviewers comment is correct, that running distance at all timepoints are not normally distributed using a Shapiro-Wilks test ($P < 0.05$). We have examined the running distances near the plateau, around week 8. We've included the frequency distribution and individual running distances at this 8 week time point below (A). With this median distance, it appears that majority of the mice align with our D2.mdx Low vs. High groupings (B). We believe that taking the total running volume throughout the intervention into consideration in our separation would allow us to address our research question in the best way.

Finally, we recognize that the absence of a WT exercise group limits our ability to make additional comparisons. However, since the main focus of our study was to evaluate the effects of volitional exercise in the dystrophic condition, we did not believe this group would be necessary. We have now acknowledged this limitation in the Methods section of our revised manuscript (pg. 5, lines 140-141), and have provided recent, impactful references that demonstrate the effects of exercise in the healthy condition. Furthermore, we have provided additional details regarding the unique characteristics of this strain, such as calcification within the muscle (pg. 5, lines 128-132).

2) The manuscript is not really focused on the threshold of mechanical activation required to have adaptative changes. The described changes in mitochondrial “function” is of interest. However, some key players are missing. For instance, pAMPK has only be measured to assess that ETA muscle of wt animals correctly respond to an acute exercise bout. In contrast, the effect of low or high VWR on the main mechano-metabolic signaling, i.e. pAMPK has not be assessed, although AMPK plays a key role in PGC-1 alpha activation and mitochondrial biogenesis. This is an important point, since most of the damaging effect of chronic forced exercise has been related to the inability of dystrophic muscle to adequately "sense" and adapt to new mechanical requirement. While I do agree that voluntary exercise may have a different outcome vs forced one and might exert benefit vs damage, the two approaches are important to actually understand the threshold of activity and need to be seen as a whole. In this regard, part of key literature is missing or not considered. For instance, basal pAMPK has been found to be higher in both C57mdx and D2 mdx, suggesting an overactivation of this signaling (already at normal cage activity) likely related to the altered mechano-transduction in dystrophin deficient muscle (Boccanegra et al DMM 2023), with minor changes exerted by treadmill exercise. I suggest to verify the state of AMPK activation in all the present conditions and to comment the results in line of the available evidences that need to be properly quoted.

We appreciate the Reviewer’s commentary regarding the altered mechano-metabolic signalling in dystrophic muscle. In our study, we collected tissues following a 48 hour washout period without access to the running wheels. Thus, we did not expect to see an acute, exercise-induced change in AMPK phosphorylation in our VWR animals. We have verified the state of AMPK activation and did not observe any difference in phosphorylation status between our D2.mdx SED and VWR animals, as anticipated.

While we agree that this altered mechanical activation in dystrophic muscle is something that should be explored in more detail, our chronic intervention is not designed to fully address this question. In our study design, we chose to remove the wheels from the cages 48 hours prior to tissue collection in order to prevent acute exercise signalling from interfering with our chronic outcomes. Thus, we are unable to explore these acute mechano-metabolic signals with our VWR protocol. To address this question, we would require a tissue collection immediately following VWR, or an acute exercise intervention. Our unpublished findings that examine the acute exercise signalling in the TA muscle following a single bout of low-to-moderate intensity exercise in D2.mdx animals show similar findings that the Reviewer presented, despite a trend towards altered ACC phosphorylation at the serine 79 site, which is downstream to AMPK, immediately post-exercise (A). Furthermore, we observe no difference in PGC-1 α protein expression in our time-course following acute exercise in D2.mdx animals (B). We hypothesize that this AMPK signalling cascade may be altered acutely following exercise in D2.mdx mice, albeit not to the same extent as in the healthy condition, which overtime, would translate to elevated PGC-1 α , as we observed after 8-10 weeks of VWR.

In support of this consideration from the Reviewer, our revised manuscript highlights the need for additional time-course studies to further investigate the altered acute mechano-metabolic signalling in dystrophic muscle (pg. 21, lines 608-611). We have also provided the additional reference that has been requested by the Reviewer.

3) The paragraph on mitochondrial respiration and the other on mitochondrial fusion needs to be revised according to the data. In many cases, no phenotypic differences are present between wt and sedentary D2mdx (see PGC 1a, OXPHOS proteins, OPA1 etc) then the change by VWR in terms of “beneficial” is not clear. By contrast, in other cases the phenotype is clear, but no effect of VWR is observed. Attention is needed to not over-interpret the results not to discuss trends in these two paragraphs and generally throughout the manuscript.

We thank the Reviewer for this comment and now recognize that these results require more clarity. We have amended our Results section to include these additional, important comparisons (pg. 15-17, lines 442-479).

4) For histology: a more direct marker of myonecrosis and inflammation should be provided. Also it is not clear if morphometric analysis has been done on the entire section (as correct) or on selected areas.

We have provided new data below that investigates the influence of VWR on inflammation and myonecrosis. Specifically, we have assessed NF- κ B activation status and IL-6R α protein expression in the GAST muscles and did not observe any differences between our D2.mdx animals (A-C). These inflammatory markers remained significantly elevated in D2.mdx mice compared to WT SED animals, while NF- κ B activation status was significantly higher in D2.mdx High VWR compared to WT SED mice. To further investigate this, we quantified inflammatory and necrotic regions within the GAST muscles across our experimental groups using a hematoxylin and eosin stain. (D, E). We found that there were significantly more necrotic and/or inflammatory regions in dystrophic muscle, with no difference between our D2.mdx SED and VWR groups. These findings suggest that VWR did not further exacerbate myonecrosis or inflammation in D2.mdx animals. In light of this, we have revised our manuscript to remove any commentary that suggests an effect of VWR on myonecrosis or inflammation. Furthermore, we performed all of our morphometric analyses on entire muscle cross-sections, and we have adjusted our Methods section to make this more clear on pg. 10-11 lines 258-260, 287-289.

5) The discussion should be revised based on the fact that a VWR of this type does not in fact help to set any specific exercise protocol. The amount of distance ran over the 24h is not a real

information on the amount of work, and important info are missing, such as continuity, fatigue, speed etc. Then I would suggest to tune down the clinical implications, as this may be misleading.

We thank the Reviewer for this feedback. We have revised our Discussion to recognize this limitation of our work (pg. 18, lines 524-526).

Other points

Keywords need to be changed to be more adherent to the aim of the work

We thank the Reviewer for this comment. We have adjusted the keywords to include “exercise” instead of “heart”.

In the key points "Volitional exercise normalized and upregulated dystrophic skeletal muscle mitochondrial respiration and content, respectively. Furthermore, a higher relative dose of VWR increased organelle fusion protein expression in D2.mdx animals": rewrite to avoid misinterpretation (higher vs what?)

We thank the Reviewer for noticing this lack of detail. We have adjusted this key point to include further detail on these comparisons to avoid misinterpretation on pg. 2, lines 35-38.

Introduction lines 99-101: "Recent literature that examined the effects of endurance-type, treadmill run training and a resistance-type synergist ablation hypertrophic model in D2.mdx mice demonstrate improved force production and reduced markers of the dystrophic pathology". The work of Hassani et al has many limitation and cannot be really compared to a physiological exercise protocol. I would suggest to take this reference out as not relevant.

We agree with this comment from the Reviewer and have removed this reference from the revised manuscript.

introduction lines 102 up to 107: "In contrast to these forced activity modalities, rodent voluntary wheel running (VWR) mimics the ability of humans to self-regulate exercise" "Further investigation of volitional exercise in DMD could contribute to the development of exercise prescriptions for patients and may uncover therapeutic avenues that modulate the dystrophic pathology" I think this statement is incorrect, the ability to self-regulate exercise can depend on the state of neuromuscular function and to other less controlled variables. As in the present study, the variability can be very high, in humans even worse. I do not think that "volitional exercise" can be translated to real exercise prescription program of physical therapy and is also a rather dangerous message, considering the high expectation of parents, that may push their DMD child to more and more volitional exercise, in spite of fatigue and in the absence of any clear indications. Then I suggest to tune this message down.

We appreciate the Reviewers feedback and have revised our Introduction section to avoid these premature clinical implications (pg. 4, lines 107-112).

Lines 415-416 "Previous research examining the effects of exercise on skeletal muscle integrity and histology in DMD are inconclusive" I think this sentence is incorrect and not necessary. Surely, there is a different effect of exercise on pathology progression based on the protocol used. But for sure is not the protocol presented herein to allow "conclusive" results.

We appreciate the Reviewers feedback and have removed this sentence from the manuscript accordingly.

Body mass for D2 wt ad mdx appear to be rather higher vs values observed in other studies. Please comment

We thank the Reviewer for this comment. We believe that our body mass values align with many previously published natural history studies that compare male, DBA/2J and D2.mdx animals at ~15-18 weeks of age (van Putten et al. FASEB J. 33(7):8110-8124, 2019; Hayes et al. Oxid Med Cell Longev. 5362115, 2022; Di Giorgio et al. Int J Mol Sci. 24(14):11805, 2023). Furthermore, the body weight of our mice appears to fall in between younger 4-10-week-old (Hughes et al. J Cachexia Sarcopenia Muscle. 10(3):643-661, 2019; Cleverdon et al. iScience. 25(9):104972), and older 21-28-week-old (Kennedy et al. Mol Ther Methods Clin Dev. 11:92-105. 2018) DBA/2J and D2.mdx animals.

Dear Dr Ljubicic,

Re: JP-RP-2024-286768R1 "Volitional exercise elicits physiological and molecular improvements in the severe D2.mdx mouse model of Duchenne muscular dystrophy" by Stephanie R Mattina, Sean Y Ng, Andrew I Mikhail, Derek W Stouth, Cora Esmenia Jornacion, Irena A Rebalka, Thomas J Hawke, and Vladimir Ljubicic

Thank you for submitting your manuscript to The Journal of Physiology. It has been assessed by a Reviewing Editor and by 2 expert referees and we are pleased to tell you that it is acceptable for publication following satisfactory revision.

REVISION CHECKLIST:

We look forward to receiving your revised submission.

Yours sincerely,

Richard Carson
Senior Editor
The Journal of Physiology

REQUIRED ITEMS

- You must start the Methods section with a paragraph headed Ethical approval (https://jp.msubmit.net/cgi-bin/main.plex?form_type=display_requirements#methods).

Research must comply with The Journal's policies regarding animal experiments (<https://physoc.onlinelibrary.wiley.com/hub/animal-experiments>) and adherence to these policies must be stated in the manuscript.

Authors should confirm in their Methods section that their experiments were carried out according to the guidelines laid down by their institution's animal welfare committee, including an ethics approval reference number. The Methods section must contain a statement about access to food, water and housing, details of the anaesthetic regime: anaesthetic used, dose and route of administration, and method of killing the experimental animals.

- Please upload separate high-quality figure files via the submission form.

- Your paper contains Supporting Information of a type that we no longer publish, including supplementary tables and figures. Any information essential to an understanding of the paper must be included as part of the main manuscript and figures. The only Supporting Information that we publish are video and audio, 3D structures, program codes and large data files. Your revised paper will be returned to you if it does not adhere to our Supporting Information Guidelines.

- Papers must comply with the Statistics Policy: https://jp.msubmit.net/cgi-bin/main.plex?form_type=display_requirements#statistics.

In summary:

- If $n \leq 30$, all data points must be plotted in the figure in a way that reveals their range and distribution. A bar graph with data points overlaid, a box and whisker plot or a violin plot (preferably with data points included) are acceptable formats.
- If $n > 30$, then the entire raw dataset must be made available either as supporting information, or hosted on a not-for-profit repository, e.g. FigShare, with access details provided in the manuscript.
- 'n' clearly defined (e.g. x cells from y slices in z animals) in the Methods. Authors should be mindful of pseudoreplication.
- All relevant 'n' values must be clearly stated in the main text, figures and tables.
- The most appropriate summary statistic (e.g. mean or median and standard deviation) must be used. Standard Error of the Mean (SEM) alone is not permitted.
- Exact p values must be stated. Authors must not use 'greater than' or 'less than'. Exact p values must be stated to three significant figures even when 'no statistical significance' is claimed.

- Please include an Abstract Figure file, as well as the Figure Legend text within the main article file. The Abstract Figure is a piece of artwork designed to give readers an immediate understanding of the research and should summarise the main conclusions. If possible, the image should be easily 'readable' from left to right or top to bottom. It should show the physiological relevance of the manuscript so readers can assess the importance and content of its findings. Abstract Figures should not merely recapitulate other figures in the manuscript. Please try to keep the diagram as simple as possible and without superfluous information that may distract from the main conclusion(s). Abstract Figures must be provided by authors no later than the revised manuscript stage and should be uploaded as a separate file during online submission labelled as File Type 'Abstract Figure'. Please also ensure that you include the figure legend in the main article file. All Abstract Figures should be created using BioRender. Authors should use The Journal's premium BioRender account to export high-resolution images. Details on how to use and access the premium account are included as part of this email.

- Please include a full title page as part of your main article (Word) file, which should contain the following: title, authors, affiliations, corresponding author name and contact details, keywords, and running title.

EDITOR COMMENTS

Reviewing Editor:

Whereas Reviewer 1 is satisfied by the revision, Reviewer 2 still has a statistical concern which should be taken in due consideration by the authors.

Senior Editor:

To echo the point raised by Referee #2 and highlighted by the reviewing editor, please make explicit any steps that were taken to ensure that the distributional properties of the data were such as to permit the use of parametric analyses. If such steps were not undertaken previously, please do so now, and alter the inferential analysis methods if appropriate.

REFEREE COMMENTS

Referee #1:

The authors have adequately addressed my concerns/questions. I have no further comments.

Referee #2:

The authors addressed diligently the criticisms raised in the rebuttal letter and in the manuscript, that appears to be improved after the revision.

One of the main concerns remains for the statistical assessment of various parameters, as data are generally presented as mean +/- SD and parametric analysis is used throughout. However, the dispersion of the data in many cases (see figs 2E,G,H; 3 G,H; 4E; 5A,B,C, J,K) raises doubts about normal distribution, as already disclosed for running distance at all timepoints (discussed in the rebuttal letter but not mentioned in the revised manuscript). The application of Shapiro-Wilks analysis should be detailed (in methods and in figure legends with results) to understand if parametric analysis is really applicable.

END OF COMMENTS

Authors' Response to Reviewer Comments:

EDITOR COMMENTS

Reviewing Editor:

Whereas Reviewer 1 is satisfied by the revision, Reviewer 2 still has a statistical concern which should be taken in due consideration by the authors.

Senior Editor:

To echo the point raised by Referee #2 and highlighted by the reviewing editor, please make explicit any steps that were taken to ensure that the distributional properties of the data were such as to permit the use of parametric analyses. If such steps were not undertaken previously, please do so now, and alter the inferential analysis methods if appropriate.

REFEREE COMMENTS

Referee #1:

The authors have adequately addressed my concerns/questions. I have no further comments.

Thank you for evaluating our paper.

Referee #2:

The authors addressed diligently the criticisms raised in the rebuttal letter and in the manuscript, that appears to be improved after the revision.

One of the main concerns remains for the statistical assessment of various parameters, as data are generally presented as mean \pm SD and parametric analysis is used throughout. However, the dispersion of the data in many cases (see figs 2E,G,H; 3 G,H; 4E; 5A,B,C, J,K) raises doubts about normal distribution, as already disclosed for running distance at all timepoints (discussed in the rebuttal letter but not mentioned in the revised manuscript). The application of Shapiro-Wilks analysis should be detailed (in methods and in figure legends with results) to understand if parametric analysis is really applicable.

We thank the Reviewer for their feedback regarding our statistical analyses. In the revised manuscript, we have now performed Shapiro-Wilks tests to assess normality in all our data. The data are mostly normally distributed. However, if our groups were not normally distributed (e.g., Fig. 2G, Fig. 2H, Fig. 5C), then we have adjusted our statistical testing and employed non-parametric Kruskal-Wallis tests followed by Dunn's *post hoc* tests accordingly. These changes

have been reflected in the Methods, Figures, and Figure Legends, and we have revised these sections to align more closely with the formatting of other articles published in the Journal of Physiology.

Dear Dr Ljubicic,

Re: JP-RP-2025-286768R2 "Volitional exercise elicits physiological and molecular improvements in the severe D2.mdx mouse model of Duchenne muscular dystrophy" by Stephanie R Mattina, Sean Y Ng, Andrew I Mikhail, Derek W Stouth, Cora Esmenia Jornacion, Irena A Rebalka, Thomas J Hawke, and Vladimir Ljubicic

We are pleased to tell you that your paper has been accepted for publication in The Journal of Physiology.

Yours sincerely,

Richard Carson
Senior Editor
The Journal of Physiology

If you would like to receive our 'Research Roundup', a monthly newsletter highlighting the cutting-edge research published in The Physiological Society's family of journals (The Journal of Physiology, Experimental Physiology, Physiological Reports, The Journal of Nutritional Physiology and The Journal of Precision Medicine: Health and Disease), please click this link, fill in your name and email address and select 'Research Roundup':
<https://www.physoc.org/journals-and-media/membernews>

- You can help your research get the attention it deserves! Check out Wiley's free Promotion Guide for best-practice recommendations for promoting your work at: www.wileyauthors.com/eoo/guide. You can learn more about Wiley Editing Services which offers professional video, design, and writing services to create shareable video abstracts, infographics, conference posters, lay summaries, and research news stories for your research at: www.wileyauthors.com/eoo/promotion.

EDITOR COMMENTS

Reviewing Editor:

Satisfactory revision. Congratulations for an interesting study.

REFEREE COMMENTS

Referee #2:

The authors adequately addressed the residual points in the revision.